# Combinatorial Bandits with Linear Constraints: Beyond Knapsacks and Fairness

**Qingsong Liu**[1*], **Weihang Xu**[1*], **Siwei Wang**[2], **Zhixuan Fang**[1,3†]

[1] IIIS, Tsinghua University  [2] Microsoft Research  [3] Shanghai Qi Zhi Institute

{liu-qs19,xuwh19}@mails.tsinghua.edu.cn
siweiwang@microsoft.com
zfang@mail.tsinghua.edu.cn

## Abstract

This paper proposes and studies for the first time the problem of combinatorial multi-armed bandits with linear long-term constraints. Our model generalizes and unifies several prominent lines of work, including bandits with fairness constraints, bandits with knapsacks (BwK), etc. We propose an upper-confidence bound LP-style algorithm for this problem, called UCB-LP, and prove that it achieves a logarithmic problem-dependent regret bound and zero constraint violations in expectation. In the special case of fairness constraints, we further provide a sharper constant regret bound for UCB-LP. Our regret bounds outperform the existing literature on BwK and bandits with fairness constraints simultaneously. We also develop another low-complexity version of UCB-LP and show that it yields $\tilde{O}(\sqrt{T})$ problem-independent regret and zero constraint violations with high-probability. Finally, we conduct numerical experiments to validate our theoretical results.

## 1 Introduction

In this paper, we study the problem of combinatorial bandits with long-term linear constraints. Our model captures important application scenarios like ad placement in online advertising systems [41], real-time traffic scheduling in wireless networks, and task assignment in crowdsourcing platforms [29], etc. Although being studied for the first time, our model subsumes several well-known problems in the Constrained Multi-Armed Bandit (CMAB) literature, including bandits with knapsacks, bandits with fairness constraints, etc. Details about these problems and how they fit into our framework are provided in Section 1.1.

Specifically, we consider an agent's online decision problem faced with a fixed finite set of $N$ arms labelled $1, 2, ..., N$, within the time horizon $T$. At each round $t$ ($1 \le t \le T$), every arm $i \in [N]$ is associated with a random reward $f_i(t) \in [0, 1]$ sampled from a time-invariant distribution $P_i$. The reward $f_i(t)$ and its distribution $P_i$ are unknown to the agent *a priori*. The mean reward of distribution $P_i$ is denoted as $\mu_i \in [0, 1]$. We denote $\boldsymbol{\mu} = (\mu_1, ..., \mu_N)^\top \in [0, 1]^N$ the mean reward vector, and define $\mu^* := \max_{i \in [N]} \mu_i$ as the maximum mean reward value of all arms. At the beginning of each round $t$, the agent is allowed to pull multiple, but no more than $m$ arms. At round $t$, the action taken by the agent is represented by an action vector $\boldsymbol{a}(t) = (a_1(t), ..., a_N(t))^\top \in \{0, 1\}^N$, where $a_i(t) = 1$ if and only if arm $i$ is pulled. The set of all feasible action vectors is defined as $\mathcal{A} = \{\boldsymbol{a} | \boldsymbol{a} \in \{0, 1\}^N, ||\boldsymbol{a}||_1 \le m\}$. After taking action $\boldsymbol{a}(t)$, the agent can observe reward from each pulled arm. Summing up the reward from the pulled arms, at round $t$, the agent receives a total reward of $R_t := \sum_{i=1}^N f_i(t)a_i(t)$.

---

[*]These authors contribute equally to this work.

[†]Corresponding author: Zhixuan Fang (zfang@mail.tsinghua.edu.cn).

36th Conference on Neural Information Processing Systems (NeurIPS 2022).

Beyond the standard combinatorial bandit setting above, we consider that the agent is subject to some constraints $\boldsymbol{g}(\cdot)$ at every round $t$, defined as $\boldsymbol{g}(\boldsymbol{a}(t)) := [g_1(\boldsymbol{a}(t)), g_2(\boldsymbol{a}(t)), ..., g_K(\boldsymbol{a}(t))]^\top$, where $g_1, g_2, \ldots, g_K : \mathbb{R}^N \to \mathbb{R}$ are linear functions. The goal of the agent is to maximize the accumulated expected reward up to the time horizon $T$, while satisfying the constraints in the long term, i.e.,

$$\max \sum_{t=1}^{T} R_t, \quad \text{s.t.} \quad \sum_{t=1}^{T} \boldsymbol{g}(\boldsymbol{a}(t)) \leq \boldsymbol{0}. \tag{1}$$

(The comparison operator $\leq$ is coordinate-wise) Define OPT(T) as the expected accumulated reward in $T$ rounds of the optimal policy satisfying the long term constraints. The agent's performance is measured in terms of regret and constraint violations defined respectively as

$$Regret_T = \text{OPT}(T) - \mathrm{E}[\sum_{t=1}^{T} R_t], \ \text{Vio}(T) = \sum_{t=1}^{T} \boldsymbol{g}(\boldsymbol{a}(t)),$$

where the expectation is taken w.r.t. the randomness of the reward and algorithm's internal randomness. Consider the following linear programming problem (LP):

$$\text{OPT}_{\text{LP}} = \max_{\boldsymbol{x} \in \mathbb{R}^N} \boldsymbol{\mu}^\top \boldsymbol{x} \quad \text{s.t.} \ \boldsymbol{g}(\boldsymbol{x}) \leq \boldsymbol{0}, \ \boldsymbol{0} \leq \boldsymbol{x} \leq \boldsymbol{1}, \ ||\boldsymbol{x}||_1 \leq m. \tag{2}$$

[4] showed that $\text{OPT}(T) \leq T \cdot \text{OPT}_{\text{LP}}$. We denote the set of optimal solutions to LP (2) as $\mathcal{X}^*$, and define $\boldsymbol{x}^* \in \mathcal{X}^*$ as one of these optimal solutions. Following a common approach, we use the optimal randomized policy $\boldsymbol{x}^*$ as the benchmark in the regret:

$$Regret_T \leq T \cdot \text{OPT}_{\text{LP}} - \mathrm{E}[\sum_{t=1}^{T} R_t] = T\langle \boldsymbol{\mu}, \boldsymbol{x}^* \rangle - \mathrm{E}[\sum_{t=1}^{T} R_t]. \tag{3}$$

## 1.1 Representative problems and related works

In this section, we outline several motivating and representative problems which fit into our general formulation and review the literature closely related to them (listed in Table 1). Problem-dependent parameters $\Delta_{\min}$ and $\Delta$ will be formally defined in Section 2.1.

Table 1: The comparison between our results and prior closely-related works. In this table, $r_{\min} = \min_i r_i$. "Single-arm" means $m = 1$. "LP" means the algorithm should solve a linear program. "Complexity" refers to the computational-complexity of the algorithm at each round.

| ALGORITHM | SETTING | ASSUMPTIONS | REGRET | VIOLATION | COMPLEXITY |
|---|---|---|---|---|---|
| [18] | SINGLE-ARM, KNAPSACKS | N/A | $O\left(\min\{N,K\}\binom{N+K}{K}\frac{\log T}{\Delta_{\min}}\right)$ | 0 | LP |
| [18] | SINGLE-ARM, KNAPSACKS | $|\mathcal{X}^*| = 1$ | $O((\min\{N,K\})^3 \log T/\Delta_{\min}^2)$ | 0 | LP |
| [42] | SINGLE-ARM, KNAPSACKS | ONE RESOURCE ($K=2$) "BEST-ARM-OPTIMALITY" | $\tilde{O}(N \log T/\bar{G}_{\text{LAG}}^2)$ | 0 | LP |
| [40] | SINGLE-ARM, KNAPSACKS | $|\mathcal{X}^*| = 1$, PRIOR KNOWLEDGE | $\tilde{O}(N \log T/\Delta_{\min})$ | 0 | LP |
| [15] | COMBINATORIAL, KNAPSACKS | ONE RESOURCE ($K=2$) | $\tilde{O}(\log^2 T)$ | 0 | LP |
| [29] | COMBINATORIAL, FAIRNESS | N/A | $O(\sqrt{mNT \log T})$ | $o(T)$ | $\tilde{O}(N)$ |
| [20] | COMBINATORIAL, FAIRNESS | N/A | $O(\sqrt{mNT \log T})$ | $o(T)$ | $\tilde{O}(N)$ |
| [48] | COMBINATORIAL, FAIRNESS | N/A | $O(\sqrt{mNT \log T})$ | $O(\sqrt{mNT \log T})$ | $\tilde{O}(N^2)$ |
| [39] | SINGLE-ARM, FAIRNESS | $\max_{i \in [N]} r_i < 1/N$ | $O(N\Delta(1 + [8\ln T/\Delta^2 - r_{\min} T]^+))$ | $O(1)$ | $\tilde{O}(N)$ |
| [12] | SINGLE-ARM, FAIRNESS | N/A | $O(N \log T/\Delta)$ | N/A | $\tilde{O}(N)$ |
| (THIS WORK) UCB-LP | COMBINATORIAL, LINEAR | N/A | $O(mN \log T/\Delta_{\min})$ | 0 | LP |
| (THIS WORK) UCB-LP | COMBINATORIAL, FAIRNESS | N/A | $O(1)$ | 0 | $\tilde{O}(N)$ |
| (THIS WORK) UCB-PLLP | COMBINATORIAL, LINEAR | N/A | $O(m\sqrt{T \log T})$ | 0 (HIGH-PROB) | $\tilde{O}(N)$ |

**Bandits with knapsacks.** The BwK (Bandits with Knapsacks) problem with deterministic costs studied in [18, 42, 40] assumes that there are $K$ resources consumed over time, each with budgets $B_1, B_2, \ldots, B_K$ respectively. Every resource $i \in [K]$ is associated with a fixed consumption vector $\boldsymbol{\lambda}_i = (\lambda_{i,1}, \lambda_{i,2}, \ldots, \lambda_{i,N}) \in \mathbb{R}_{\geq 0}^N$. The agent maximizes the accumulated reward subject to the budget constraints $\sum_{i \in [T]} \boldsymbol{\lambda}_i^\top \boldsymbol{a}(t) \leq B_i, \forall i \in [K]$. One can see that, this problem is equivalent with the special case of our setting where the linear constraint functions have the form $\boldsymbol{g}(\boldsymbol{a}(t)) = (\boldsymbol{\lambda}_1^\top \boldsymbol{a}(t) - B_1/T, \ldots, \boldsymbol{\lambda}_K^\top \boldsymbol{a}(t) - B_K/T)$.

The existing BwK literature [1, 4, 17] has derived algorithms that achieve the problem-independent regret bounds of the similar order $\tilde{O}(\sqrt{KNT})$. Such order of the regret bound is also obtained

by [41, 42, 28] for combinatorial setting. However, the problem-dependent regret of BwK with deterministic costs is less explored. [18] achieved one regret bound of $O\left(\min\{N,K\}\binom{N+K}{K}\frac{\log T}{\Delta_{\min}}\right)$ with unsatisfying exponential dependence on $K$ and $N$. They also provide another regret bound of $O((\min\{N,K\})^3\frac{\log T}{\Delta_{\min}^2})$ with polynomial dependence on $N, K$ but require an additional assumption of the optimal solution of LP (2) being unique. Later [40] improves this regret to $O(N\frac{\log T}{\Delta_{\min}})$ also under the assumption that LP (2) has an unique optimal solution. But their algorithm requires the knowledge of some parameters of the problem instance a priori (characterized by $\Delta_{\min}$ and $\mu^*$), which is unpractical. With the "best-arm-optimality" assumption, i.e., there is an optimal policy that only pulls one arm, [42] achieved the regret of $O(N \cdot \log T/G_{\mathrm{LAG}}^2)$ ($G_{\mathrm{LAG}}$ is their defined Lagrangian gap) when there is only one resource ($K = 2$), whose practicability is also restricted. Very recently, [15] derives a regret bound of $O(\log^2 T)$ for combinatorial setting and one resource constraint, which is the best so far in the area of combinatorial BwK but still sub-optimal in terms of $T$.

For BwK problems with stochastic costs, [18, 47, 46, 42, 30, 7] obtained logarithmic regrets under different restrictive assumptions, e.g., non-degeneracy of (2), $K = 2$ (single source) or "best-arm-optimality". [18, 42] showed that it is impossible to achieve any problem-dependent regret bound of $o(\sqrt{T})$ without additional assumptions in general.

**Bandits with fairness constraints.** Recently, [29, 48, 39] studied the problem of bandits with fairness constraints. Under their setting, the agent maximizes the cumulative expected reward, and needs to ensure that each arm $i \in [N]$ is pulled for at least $r_i \in (0,1)$ fraction of times at the end of $T$ rounds. In our model, such fairness constraints are equivalent with the following special kind of linear long-term constraints: $\boldsymbol{g}(\boldsymbol{a}(t)) = -\boldsymbol{a}(t) + \boldsymbol{r}$, where $\boldsymbol{r} = (r_1, r_2, \ldots, r_N)^\top$.

[29] first studied the combinatorial (sleeping) bandits with fairness constraints. Their algorithm LFG combines virtual queue technique and UCB learning. LFG yields $\tilde{O}(\sqrt{T})$ regret and sublinear ($o(T)$) constraint violations. [20] replaced the UCB learning with Thompson Sampling in LFG and obtained performance guarantees with the same order. Later [48] improved the constraint violations bound to $O(\sqrt{T})$ with the same regret order by using online convex optimization techniques and RRS rounding. A big advancement is made by recent works [12, 39] that they achieve $O(\log T)$ and $o(\log T)$ regret bounds, respectively, for MAB ($m = 1$) with fairness constraints based on the modified UCB1 algorithm. And [16] achieved a "penalized" regret bound of $O(\log T)$ in the single-arm setting.

**Bandits with group fairness.** In scenarios like ad-display optimization, the agent is subject to the group fairness constraint, e.g., arms belonging to one group should be pulled more frequently than arms belonging to another group [36], or the arm with higher average reward should be pulled more times than the arm with lower average reward [23], etc. This problem also fits into our formulation of linear constraints. Only problem-dependent regrets of $\tilde{O}(\sqrt{T})$ are obtained in related works.

**Other related literature.** A large body of literature (e.g., [10, 45]) derived $O(\log T)$ regret bounds for unconstrained combinatorial bandits. [22, 24] studied the BwK problem under the adversarial setting. [2, 38, 3, 35] studied constrained linear bandits. Our framework is also related to online convex optimization with long-term constraints (e.g., [33, 11, 51, 50]), where the agent faces several convex constraints and these constraints need to be satisfied in the long term. We remark that, in this setting, the full reward function at each round would be revealed after the decision making, which is in contrast to our setting that we do not have such observation due to the semi-bandit feedback.

## 1.2 Discussion: significance of linear constraints

In this section, we discuss the significance of generalization to linear constraints. Previous literature on constrained bandits only studies constraints with specific forms, namely, the fairness constraint and the knapsacks constraint, etc. As we only require $\boldsymbol{g}(\cdot)$ to be linear, our model not only subsumes both of them as its special cases, but also solves a larger group of constraints, enabling broader applications. For example: **1.** Our model for the first time addresses scenarios where both fairness and knapsack constraints exists simultaneously. **2.** Our model solves the case of weighted fairness constraints, i.e., weighted sums of pulls of each arm are required to be larger than a threshold: $\sum_{i=1}^{N} \kappa_{ij} h_i(T) \geq r_j T$, where $\kappa_{ij}$ is the $j^{\mathrm{th}}$ weight of the $i^{\mathrm{th}}$ arm. In contrast, traditional fairness constraints treat each arm as unweighted and could be seen as a special case of weight fairness constraints.

In fact, there are many applications where combinatorial bandits with various complicated linear constraints are required. We show some concrete examples as follows. (a) In crowdsourcing, where a

group of workers are assigned with tasks to achieve high accuracy, a set of complex linear constraints occurs: each tasks should have enough workers, while each worker should have a fair workload. (b) In network routing where multiple paths (each path consists of a series of links) needs to be selected to send the traffic within the time budget and the bandwidth constraints. (c) Case of Internet of Things where a set of sensors need to be selected at each round to guarantee the QoS requirement [21] (e.g., throughout, mean-delay) under budget constraint on data collection cost (e.g., energy consumption).

### 1.3 Our contributions

Our main contributions are summarized below.

(a) We define a general formulation termed combinatorial bandits with linear long-term constraints. In contrast to previously outlined pieces of work, we consider the problem in its full generality and do not assume or require any prior knowledge of the problem instance. We design an upper-confidence bound LP-style algorithm named UCB-LP, and develop a novel analytical technique to build a relationship between LP solution (distribution support over arms) and "reward allocation", with which we show that UCB-LP guarantees a problem-dependent regret of $O(mN\frac{\log T}{\Delta_{\min}})$ and no constraint violation in expectation. To the best of our knowledge, this is the first logarithmic regret bound for bandits with long term linear constraints under the combinatorial setting.

(b) For the special case of fairness constraints, we show that UCB-LP achieves $O(1)$ regret and guarantees zero expected constraint violation at the same time. To the best of our knowledge, both the regret and the constraint violations outperform all existing works on combinatorial bandits with fairness constraints. We also show that UCB-LP has a low running time of $\tilde{O}(N)$ in this special case.

(c) To overcome the potentially time-consuming LP in UCB-LP, we further develop a low-complexity version of UCB-LP, called UCB-PLLP, since it builds on the Lagrangian (L) of UCB-LP and pessimistically (P) tracks the constraint violations. We show that it yields $\tilde{O}(\sqrt{T})$ problem-independent regret and guarantees zero constraint violations for any $\tau \leq T$ with high-probability. The computational complexity of UCB-PLLP is $\tilde{O}(N)$.

## 2 Main results

In this section, we present our algorithm and corresponding performance analysis for our general formulation. All the proofs of listed lemmas, propositions and corollaries are deferred to the supplementary material. Our experimental results are given in the suppmetary material.

### 2.1 Preliminary: notations and existing techniques

**Notations.** For every arm $i$, define $h_i(t) := \sum_{\tau=1}^{t-1} a_i(\tau)$ as the number of pulls of it at the beginning of round $t$, and $\bar{\mu}_i(t) := \frac{1}{h_i(t)}\sum_{\tau=1}^{t-1} a_i(\tau)f_i(\tau)$ as its empirical reward estimate at round $t$. Denote the feasible region of LP (2) as $\mathcal{D} := \{\boldsymbol{x}|\boldsymbol{x} \in [0,1]^N, ||\boldsymbol{x}||_1 \leq m, \boldsymbol{g}(\boldsymbol{x}) \leq \boldsymbol{0}\}$. In this paper, for any vector $\boldsymbol{v}$, we use $v_i$ to denote its $i^{\text{th}}$ coordinate. For any event $E$, we use $\overline{E}$ to denote its negation.

**Extreme points and general sub-optimality measure.** Note that the feasible region $\mathcal{D}$ to LP (2) is a convex polytope, and we let $\mathcal{B}$ be the set of its extreme points. An extreme point of $\mathcal{D}$ is a point in $\mathcal{D}$ which does not lie in any open line segment joining two points of $\mathcal{D}$. It is well-known in the theory of LP that extreme points and basic feasible solutions are equivalent, and any LP attains its optimal value at an extreme point [6]. Recall that $\boldsymbol{x}^*$ is an optimal solution to LP (2), and we define the sub-optimality gap for any $\boldsymbol{x} \in \mathcal{B}$ as $\Delta_{\boldsymbol{x}} := \langle \boldsymbol{\mu}, \boldsymbol{x}^* \rangle - \langle \boldsymbol{\mu}, \boldsymbol{x} \rangle$. Define $\Delta_{\min} := \min_{\boldsymbol{x} \in \mathcal{B} \setminus \mathcal{X}^*} \Delta_{\boldsymbol{x}}$. Since $\mathcal{B}$ is finite, $\Delta_{\min}$ is well-defined and strictly positive. The same definition of the sub-optimality gap is also used in [18]. We note that $\Delta_{\min}$ can be seen as a generalization of the minimum sub-optimality gap in standard MAB problem defined as $\Delta := \min_{i:\mu_i \neq \mu^*} |\mu^* - \mu_i|$. Specifically, under the standard MAB setting, $m = 1, \mathcal{B} = \{\boldsymbol{a}|\boldsymbol{a} \in \{0,1\}^N, ||\boldsymbol{a}||_1 \leq 1\}$, $\Delta_{\min}$ coincides with $\Delta$. In this paper, we will state our problem-dependent regret bounds in terms of $\Delta_{\min}$.

### 2.2 The general algorithm UCB-LP and its performance analysis

Now we introduce our algorithm UCB-LP for combinatorial bandits with long-term linear constraints. UCB-LP is a generalization of SemiBwK algorithm [41] to the general linear constraints setting which

chooses arms through randomized policy. In our setting, one main challenge is that no super-arm is optimal across all rounds, but there exists an optimal sampling distribution over arms and the intuition behind UCB-LP is to identify such distribution and sampling arms based on it. At each round, UCB-LP consists of two stages. In the first stage, we first compute the truncated UCB estimate vector $\hat{\boldsymbol{\mu}}(t) \in \mathbb{R}^N$ defined as $\hat{\mu}_i(t) = \min\left\{\bar{\mu}_i(t) + \sqrt{\frac{2\ln t}{h_i(t)}}, 1\right\}, \forall i \in [N]$. Then we solve the following LP and get an optimal solution $\boldsymbol{x}(t) \in \mathcal{B}$:

$$\max_{\boldsymbol{x} \in \mathbb{R}^N} \ \langle \hat{\boldsymbol{\mu}}(t), \boldsymbol{x} \rangle \qquad \text{s.t.} \quad \boldsymbol{g}(\boldsymbol{x}) \leq \boldsymbol{0}, \ \boldsymbol{0} \leq \boldsymbol{x} \leq \boldsymbol{1}, \ ||\boldsymbol{x}||_1 \leq m. \tag{4}$$

(Note that LP (2) and LP (4) have the same feasible region $\mathcal{D}$, hence the same set of extreme points $\mathcal{B}$.) Here we require the solved optimal solution $\boldsymbol{x}(t)$ to be an extreme point of LP (4), i.e., $\boldsymbol{x}(t) \in \mathcal{B}$. In fact, this is naturally satisfied by many LP algorithms, e.g., Simplex Method. Even if the optimal solution of LP is not an extreme point, it can be efficiently converted to one by tightening some slack constraints in (4) [37, 19].

The second stage is to construct a distribution $\pi_t(\cdot)$ over $\mathcal{A}$ with expectation $\boldsymbol{x}(t)$, i.e., $\sum_{\boldsymbol{a} \in \mathcal{A}} \pi_t(\boldsymbol{a}) \cdot \boldsymbol{a} = \boldsymbol{x}(t)$, and sample super-arm $\boldsymbol{a}(t) \sim \pi_t$. Since $\boldsymbol{x}(t) \in \mathcal{D} \subseteq \text{Conv}(\mathcal{A})$, such $\pi_t$ exists and can be generated via a convex decomposition of $\boldsymbol{x}(t)$. Although there are many randomized rounding methods with $O(N^2)$ running time to achieve this, we show in the supplementary material that computing $\pi_t$ and sampling $\boldsymbol{a}(t)$ can be finished in $O(N \log N)$ time. The idea of sampling arms maintaining the marginal distribution are also used in [13, 52]. Finally, we pull the arms according to $\boldsymbol{a}(t)$, observe the reward value of pulled arms, and update the statistics.

Although motivated by SemiBwK algorithm [41], UCB-LP deals with general linear constraint function without any assumptions and our goal is to derive a problem-dependent regret bound. Therefore, new techniques have to be developed for the analysis of UCB-LP.

The following theorem provides the generic bounds of regret and constraint violations for UCB-LP.

**Theorem 1** *UCB-LP satisfies*

$$Regret_T = O\left(\frac{mN \log T}{\Delta_{\min}}\right), \tag{5}$$

$$E[\text{Vio}(T)] \leq \boldsymbol{0}. \tag{6}$$

**Proof sketch of Theorem 1:** Note that (6) is straightforward from LP (4) and the fact that $\boldsymbol{g}(\cdot)$ is linear. Now we present the main idea of proving (5). To obtain the regret bound, we cannot directly apply the traditional analysis from bandit community here as we cannot bound $h_i(t)$ and the algorithm might favor sub-optimal arms even if they have already been pulled for $\Omega(\log T)$ times due to the structural property of the LP. Instead, we bound the number of times UCB-LP fails to yield the optimal policy, i.e., $\boldsymbol{x}(t) \notin \mathcal{X}^*$. The most natural idea to achieve this is to bound the number of times $\boldsymbol{x}(t) = \overline{\boldsymbol{x}}$ for every sub-optimal policy $\overline{\boldsymbol{x}} \in \mathcal{B} \setminus \mathcal{X}^*$. However, such an idea fails in the sense that the resulting bound would scale with $|\mathcal{B}|$, which is exponentially large. Addressing this technical challenge is nontrivial and previous works circumvented this problem by assuming special structures (e.g., [18] and [40] assumed $|\mathcal{X}^*| = 1$, [42] assumed only one resource and $||\boldsymbol{x}^*||_0 = 1$, etc) or using the prior knowledge of some problem-dependent parameters [40]. In our proof, we overcome this difficulty directly by handling the case $\boldsymbol{x}(t) \notin \mathcal{X}^*$ with the idea of "regret allocation".

Define $\boldsymbol{w}(t) = (\sqrt{\frac{2\ln t}{h_i(t)}}, .., \sqrt{\frac{2\ln t}{h_N(t)}})^\top$. Since the regret only occurs when $\boldsymbol{x} \notin \mathcal{X}^*$, we claim that $\langle \boldsymbol{\mu}, \boldsymbol{x}^* \rangle \leq \langle \hat{\boldsymbol{\mu}}(t), \boldsymbol{x}(t) \rangle$, $\langle \hat{\boldsymbol{\mu}}(t), \boldsymbol{x}(t) \geq \langle \boldsymbol{\mu}(t), \boldsymbol{x}(t) \rangle + \Delta_{\boldsymbol{x}(t)}$ and $\langle \boldsymbol{w}, \boldsymbol{x}(t) \rangle \geq \Delta_{\boldsymbol{x}(t)}/2$ all hold with high probability in such case since $\langle \hat{\boldsymbol{\mu}}(t), \boldsymbol{x}(t) \rangle \geq$ (LP property)$\langle \hat{\boldsymbol{\mu}}(t), \boldsymbol{x}^* \rangle \geq$ (high-prob)$\langle \boldsymbol{\mu}, \boldsymbol{x}^* \rangle \geq \langle \boldsymbol{\mu}, \boldsymbol{x}(t) \rangle + \Delta_{\boldsymbol{x}(t)} \Rightarrow \langle \hat{\boldsymbol{\mu}}(t) - \boldsymbol{\mu}, \boldsymbol{x}(t) \rangle \geq$ (high-prob)$\Delta_{\boldsymbol{x}(t)} \Rightarrow \langle 2\boldsymbol{w}, \boldsymbol{x}(t) \rangle \geq$ (high-prob)$\Delta_{\boldsymbol{x}(t)} \Rightarrow \langle \boldsymbol{w}, \boldsymbol{x}(t) \rangle \geq$ (high-prob)$\Delta_{\boldsymbol{x}(t)}/2$. With these properties, when $\boldsymbol{x} \notin \mathcal{X}^*$, we could allocate the incurred regret $R_t = E[\langle \boldsymbol{\mu}, \boldsymbol{x}^* \rangle - \langle \boldsymbol{\mu}, \boldsymbol{x}(t) \rangle] = \Delta_{\boldsymbol{x}(t)}$ to every base arm in the following way, and we will argue that this allocation is correct since:

$$R_t = E[\langle \boldsymbol{\mu}, \boldsymbol{x}^* \rangle - \langle \boldsymbol{\mu}, \boldsymbol{x}(t) \rangle] = \Delta_{\boldsymbol{x}(t)} = E[\langle \hat{\boldsymbol{\mu}}(t) - \boldsymbol{\mu}, \boldsymbol{x}(t) \rangle] - E[\langle \boldsymbol{\mu}, \boldsymbol{x}^* \rangle - \langle \hat{\boldsymbol{\mu}}(t), \boldsymbol{x}(t) \rangle]$$

$$\leq \text{(high-prob)} E[\langle \hat{\boldsymbol{\mu}}(t) - \boldsymbol{\mu}, \boldsymbol{x}(t) \rangle] \leq \text{(high-prob)} E[\langle 2\boldsymbol{w}, \boldsymbol{x}(t) \rangle] = 2E\left[\sum_{i \in [N]} x_i(t)\sqrt{2\ln t/h_i(t)}\right]$$

Thus each arm contributes $2x_i(t)\sqrt{\frac{2\ln t}{h_i(t)}}$ to the regret $\Delta_{\boldsymbol{x}(t)}$. Next we claim that the sum above regret allocation on all arms is dominated by the arms in the set $V(t) = \{i : h_i(t) \leq \frac{32m^2 \ln t}{\Delta_{\min}^2}\}$, i.e., $\frac{1}{2}\sum_{i \in [N]} 2x_i(t)\sqrt{\frac{2\ln t}{h_i(t)}} \leq \sum_{i \in V(t)} 2x_i(t)\sqrt{\frac{2\ln t}{h_i(t)}}$, which is derived by:

$\sum_{i \in [N]/V(t)} x_i(t)\sqrt{\frac{2\ln t}{h_i(t)}} \le \sum_{i \in [N]/V(t)} x_i(t)\frac{\Delta_{\min}}{4m} \le \sum_{i \in [N]} x_i(t)\frac{\Delta_{\min}}{4m} \le \frac{\Delta_{\min}}{4} \le \frac{\Delta_{\boldsymbol{x}(t)}}{4}$, and
$\sum_{i \in [N]} x_i(t)\sqrt{\frac{2\ln t}{h_i(t)}} = \langle \boldsymbol{w}(t), \boldsymbol{x}(t)\rangle \ge \frac{\Delta_{\boldsymbol{x}(t)}}{2}$.

Therefore, the following total regret decomposition holds with high-probability:

$$Regret_T \le 2E[\sum_{t=1}^{T}\sum_{i \in [N]} x_i(t)\sqrt{2\ln t/h_i(t)}\mathbf{I}\{\boldsymbol{x}(t) \notin \mathcal{X}^*\}]$$

$$\le 4E[\sum_{t=1}^{T}\sum_{i \in V(t)} x_i(t)\sqrt{2\ln t/h_i(t)}\mathbf{I}\{\boldsymbol{x}(t) \notin \mathcal{X}^*\}] \le 4E[\sum_{t=1}^{T}\sum_{i \in V(t)} x_i(t)\sqrt{2\ln t/h_i(t)}]. \tag{7}$$

Define $G(i) = \{t : i \in V(t)\}$, and $T_i = \arg\max_{\tau \in G(i)} \tau$. Continuing from (7) we obtain

$$Regret_T \le 4E[\sum_{t=1}^{T}\sum_{i \in V(t)} x_i(t)\sqrt{2\ln t/h_i(t)}] = 4E[\sum_{i \in N}\sum_{t \in G(i)} x_i(t)\sqrt{\frac{2\ln t}{h_i(t)}}] \le 4E[\sum_{i \in N}\sum_{t=1}^{T_i} x_i(t)\sqrt{\frac{2\ln t}{h_i(t)}}].$$

The remaining problem is to bound $4E[\sum_{i \in N}\sum_{t=1}^{T_i} x_i(t)\sqrt{\frac{2\ln t}{h_i(t)}}]$, which is solved by the following lemma.

**Lemma 1** $E[\sum_{\tau=1}^{t} x_i(\tau)\sqrt{\ln \tau/h_i(\tau)}] \le 3\sqrt{\ln t} \cdot E[\sqrt{h_i(t) + 1}]$.

Applying Lemma 1 yields $4E[\sum_{i \in N}\sum_{t=1}^{T_i} x_i(t)\sqrt{\frac{2\ln t}{h_i(t)}}] \le 12\sum_{i \in [N]}\sqrt{2\ln T_i(h_i(T_i) + 1)} \le 12\sum_{i \in [N]}\sqrt{2\ln T_i(\frac{32m^2\ln T_i}{\Delta_{\min}^2} + 1)}$ (Since $T_i \in V(T_i)$) $\le 12\sum_{i \in [N]}\sqrt{2\ln T(32m^2\ln T/\Delta_{\min}^2 + 1)} = O(\frac{mN}{\Delta_{\min}}\ln T)$. Putting all things together completes the proof of Theorem 1.

**Proof of Lemma 1:** Our proof idea is using the facts (a): $1/\sqrt{x} \le 3(\sqrt{x+1} - \sqrt{x})$, $x \ge 1$; (b): $E[\sqrt{h_i(\tau + 1)}|\mathcal{F}_t] = E[x_i(\tau)\sqrt{h_i(\tau) + 1}|\mathcal{F}_{\tau-1}] + E[(1 - x_i(\tau))\sqrt{h_i(\tau)}|\mathcal{F}_{\tau-1}]$, where $\mathcal{F}_t = \{\boldsymbol{x}(\tau)\}_{\tau=1}^{t}$. Then $E[\sum_{\tau=1}^{t} E[x_i(\tau)\sqrt{\ln \tau/h_i(\tau)}|\mathcal{F}_{\tau-1}]] \le \sqrt{\ln t}E[\sum_{\tau=1}^{t} E[x_i(\tau)\sqrt{1/h_i(\tau)}|\mathcal{F}_{\tau-1}]] \overset{(a)}{\le} 3\sqrt{\ln t}E[\sum_{\tau=1}^{t} E[x_i(\tau)(\sqrt{h_i(\tau) + 1} - \sqrt{h_i(\tau)})|\mathcal{F}_{\tau-1}] \overset{(b)}{=} 3\sqrt{\ln t}E[\sum_{\tau=1}^{t} E[\sqrt{h_i(\tau + 1)}|\mathcal{F}_\tau] - E[\sqrt{h_i(\tau)}|\mathcal{F}_{\tau-1}]] \le 3\sqrt{\ln t}E[E[\sqrt{h_i(t + 1)}|\mathcal{F}_t]] \le \sqrt{\ln t}E[\sqrt{h_i(t + 1)}] \le 3\sqrt{\ln t}E[\sqrt{h_i(t) + 1}]$.

**Comparison with previous results.** Several previous works (e.g., [18, 42, 40, 39]) studying bandits with long term constraints have also achieved $O(\log T)$ regret bounds. Our regret bound has four major improvements over theirs: (a) Our regret bound has a better dependence on $\Delta_{\min}$ and $N$. (b) Previous regret bounds only apply to the single-armed setting, while ours applies to the combinatorial setting. (c) Our regret bound is valid for all linear constraints, while theirs is only valid for specific kind of constraints, either BwK with deterministic costs, or fairness constraints (which are the special cases of linear constraints). (d) Almost all of them require additional assumptions or knowledge of some parameters of the problem instance a prior. For example, [18] explicitly admitted that their (poly-)logarithmic regret and the corresponding analysis only valid under the assumption of $|\mathcal{X}^*| = 1$. (See Appendix A.1 and Assumption 9 in their arxiv version.) However, $|\mathcal{X}^*| = 1$ does not always hold. When consider BwK problem with only one resource, if there exist arms $p$, $q$ such that $\frac{\mu_p}{\lambda_p} = \frac{\mu_q}{\lambda_q}$, where $\lambda_i$ is the amount of resource consumed by arm $i$ if it gets pulled, then LP (2) has non-unique optimal solution, i.e., $|\mathcal{X}^*| \ne 1$. Even consider problem of bandits with fairness constraints, if there exist two arms with the same required probability of being pulled, the optimal solution to LP (2) also may not be unique. Beyond assuming $|\mathcal{X}^*| = 1$, the algorithms in [40] still require some prior knowledge of the problem instance (characterized by $\mu^*$ and $\Delta_{\min}$). Substantial assumptions including only one resource and "best-arm-optimality" are also required in [42]. On the contrary, we do not require any assumptions and prior knowledge of the problem instance.

**Reduction to unconstrained (combinatorial) bandits setting.** When there are no constraints, i.e., $\boldsymbol{g}(t) = \boldsymbol{0}$, UCB-LP reduces to the standard (Comb) UCB1 algorithm [25] and we could also recover the results of [25] in such case. This further justifies the tightness of our regret bound.

## 2.3 Achieving constant regret for fairness constraints

In this section, we consider the special case of fairness constraints. Following the same setting in the literature, e.g., [29, 48, 39], each arm $i \in [N]$ is required to be pulled at least $r_i \in (0, 1)$

fraction of times, and $\sum_{i \in [N]} r_i < m$. Namely, the agent has to satisfy $h_i(T) \geq r_i T, \forall i \in [N]$. Define $\boldsymbol{r} = (r_1, r_2, \ldots, r_N)^\top$, then the equivalent linear long term constraint function $\boldsymbol{g}(\cdot)$ under our setting is $\boldsymbol{g}(\boldsymbol{a}(t)) = -\boldsymbol{a}(t) + \boldsymbol{r}$. In the fairness constraints setting, the main challenge is that the time horizon $T$ is unknown to the algorithm beforehand, and thus the "forced exploration" trick cannot be directly applied here. A lot of work (See Table 1) has sprung up recently to tackle this difficulty, but the best previous result is only $\tilde{O}(\sqrt{T})$ under the combinatorial setting. The following theorem shows that UCB-LP could guarantee a constant regret.

**Theorem 2** *Define $r_{\min} := \min_{i \in [N]} r_i$. In the case of fairness constraints, UCB-LP guarantees that*
$$Regret_T \leq \frac{32mN^2}{r_{\min}^2 \Delta_{\min}^2} \ln^2 \left( \frac{32N^2}{r_{\min}^2 \Delta_{\min}^2} \right) + \frac{mN\pi^2}{2}.$$

**Basic analysis of Theorem 2.** To derive a constant regret, one natural idea is to show that the quantity $E[h_i(T)] - r_i \cdot T$ is bounded for every sub-optimal arm $i$ (whose mean reward is not Top-$m$). However, this is not the case in classic bandit analysis (e.g., for UCB-based algorithms) and thus cannot directly apply here. Our main proof idea is to show that UCB-LP only chooses a sub-optimal distribution over arms (i.e., $\boldsymbol{x}(t) \notin \mathcal{X}^*$) a limited number of times for fairness constraints. To achieve this, we first transform the event of $\boldsymbol{x}(t) \notin \mathcal{X}^*$ to the events associated with arm's UCB estimate error, i.e., we use the following lemma to characterize the case of $\boldsymbol{x}(t) \notin \mathcal{X}^*$ by the sensitivity analysis of LP (4).

**Lemma 2** *If $||\hat{\boldsymbol{\mu}}(t) - \boldsymbol{\mu}||_1 < \Delta_{\min}$, then $\boldsymbol{x}(t) \in \mathcal{X}^*$.*

In other words, when $\boldsymbol{x}(t) \notin \mathcal{X}^*$ we have $||\hat{\boldsymbol{\mu}}(t) - \boldsymbol{\mu}||_1 > \Delta_{\min}$. And then we just have to prove that $||\hat{\boldsymbol{\mu}}(t) - \boldsymbol{\mu}||_1 > \Delta_{\min} \Rightarrow \max_{i \in [N]} |\hat{\mu}_i(t) - \mu_i| \geq \Delta_{\min}/N$ only happens a finite number of times. This is a little counterintuitive for UCB-style algorithms, but it holds for UCB-LP in the fairness setting as UCB-LP guarantees that every arm has a positive probability of being pulled at each round which leads to the diminishing confidence width of all arms and the bonus of concentration inequality. In particular, by martingale analysis and extending the concentration bound to a random process that evolves over time, we prove that there exists a constant $c > 0$ (e.g., $\frac{32N^2}{r_{\min}^2 \Delta_{\min}^2} \ln^2 \frac{32N^2}{r_{\min}^2 \Delta_{\min}^2}$) such that $h_i(t) \geq \frac{8N^2}{\Delta_{\min}^2} \ln t$ holds with high-probability for each arm $i$ when $t > c$, which gives that $|\hat{\mu}_i(t) - \mu_i| \leq \Delta_{\min}/N$ holds with high-probability when $t > c$. It is worth noting that such properties do not hold for classic bandit algorithms (e.g., algorithms based on the UCB1 framework). And to the best of our knowledge there is no such results in the literature. Finally, to fit into our Lemma 2, we handle the regret as $Regret_T \leq m \sum_{t=1}^{T} \Pr[\boldsymbol{x}(t) \notin \mathcal{X}^*] \leq mc + m \sum_{t=c+1}^{T} \Pr[\boldsymbol{x}(t) \notin \mathcal{X}^*] \leq mc + m \sum_{t=c+1}^{T} \Pr[||\hat{\boldsymbol{\mu}}(t) - \boldsymbol{\mu}||_1 \geq \Delta_{\min}]$. Put all things together then we complete the proof.

**Breaking the $\Omega(\log T)$ lower bound.** The constant regret bound given by Theorem 2 might seem counter-intuitive at first glance, since the lower bound of regret under classical MAB setting is $\Omega(\log T)$ [26]. However, note that the fairness level $r_i$ is required to be strictly positive, and the benchmark policy in the regret also needs to satisfy the long term constraints. This distinguishes our setting from the classical MAB setting. Thus, the traditional $\Omega(\log T)$ lower bound no longer applies.

The reason why UCB-LP achieves a constant regret is: UCB-LP guarantees that every arm has a positive probability of being pulled at each round. This leads to more accurate reward estimates of arms and a diminishing gap between $\hat{\boldsymbol{\mu}}(t)$ and $\boldsymbol{\mu}$, which makes UCB-LP output the sub-optimal action distribution only a finite number of times. That said, it still requires non-trivial techniques to establish a constant regret bound. In fact, even $O(\log T)$ regret bound has not yet been achieved for combinatorial setting in prior works.

Note that our regret bound has a quadratic dependence on $1/r_{\min}$, which goes to infinity as $\boldsymbol{r} \to \boldsymbol{0}$. In fact, when $\boldsymbol{r} = \boldsymbol{0}$, i.e., there are no fairness constraints, our problem degenerates to classical MAB setting where achieving constant regret bound is impossible. This indicates that such dependence on extra parameters is inevitable for any constant regret bound. Of course, the general $O(\log T)$ regret bound developed in Theorem 1 still applies when $\boldsymbol{r} \to \boldsymbol{0}$, since it does not depend on $1/r_{\min}$.

$\tilde{O}(N)$ **running time.** The structural property of the fairness constraints allows us to obtain a closed form solution to LP (4), as shown in the following proposition.

**Proposition 1** *For each round $t$, rearrange the coordinates of UCB estimate vector $\hat{\boldsymbol{\mu}}(t)$ in descending order such that $\hat{\mu}_{\sigma_1^t}(t) \geq \hat{\mu}_{\sigma_2^t}(t) \geq \cdots \geq \hat{\mu}_{\sigma_N^t}(t)$, where $\sigma_1^t, \ldots, \sigma_N^t$ is a permutation of $1, \ldots, N$. Then one of the optimal extreme points to LP (4) (denoted as $\boldsymbol{x}(t)$) has the following form:*

$$x_{\sigma_i^t}(t) = 1, \; \forall i < k_t; \; x_{\sigma_i^t}(t) = r_{\sigma_i^t}, \forall i > k_t; x_{\sigma_{k_t}^t}(t) = m + 1 - k_t - \sum_{i > k_t} r_{\sigma_i^t},$$

*where* $k_t = \min\{q \in \mathbb{Z}^+ \mid \sum_{i=1}^q (1 - r_{\sigma_i^t}) \geq m - \sum_{i=1}^N r_i\}$.

Proposition 1 immediately implies that UCB-LP is computationally efficient when dealing with fairness constraints. To solve LP (4), one can simply sort all coordinates of $\hat{\boldsymbol{\mu}}(t)$ in $O(N \log N)$ time, then compute $k_t$ and $\boldsymbol{x}(t)$ according to the closed form expression in proposition 1 (note that $\boldsymbol{x}(t)$ in Proposition 1 also lies in $\mathcal{B}$). Since the running time of computing distribution $\pi_t$ and sampling $\boldsymbol{a}(t)$ is also $O(N \log N)$, we conclude that the time average complexity of UCB-LP is $O(N \log N) = \tilde{O}(N)$.

**Another constant regret bound.** Proposition 1 further provides two important implications:

(a) Since LP (2) has the same form with LP (4), the optimal solution $\boldsymbol{x}^*$ to LP (2) can also be written in closed form in the same manner to Proposition 1. Specifically, rearrange the coordinates of $\boldsymbol{\mu}$ in descending order as $\mu_{\sigma_1} \geq \mu_{\sigma_2} \geq \cdots \geq \mu_{\sigma_N}$, and define $k = \min\{q \in \mathbb{Z}^+ \mid \sum_{i=1}^q (1 - r_{\sigma_i}) \geq m - \sum_{i=1}^N r_i\}$. Then $\boldsymbol{x}^*$ has the following form:

$$x_{\sigma_i}^* = 1, \; \forall i < k; \; x_{\sigma_i}^* = r_{\sigma_i}, \forall i > k; x_{\sigma_k}^* = m + 1 - k - \sum_{i > k} r_{\sigma_i}.$$

(b) The optimal solution $\boldsymbol{x}(t)$ to LP (4) only depends on the relative order, not the absolute value of $\hat{\mu}_1(t), \ldots, \hat{\mu}_N(t)$. This motivates us to characterize the sensitivity of LP (4) from a new perspective. Intuitively, if $\hat{\boldsymbol{\mu}}$ and $\boldsymbol{\mu}$ are close enough such that the relative order between the coordinates is (at least partially) preserved, then the optimal solutions of LP (4) and LP (2) will coincide.

The above two implications motivate us to define a new parameter $\epsilon := \min_{\mu_i \neq \mu_{\sigma_k}} |\mu_i - \mu_{\sigma_k}|$, and propose the following corollary to characterize the sensitivity of LP (4).

**Corollary 1** *If* $||\hat{\boldsymbol{\mu}}(t) - \boldsymbol{\mu}||_\infty < \frac{\epsilon}{2}$, *then* $\boldsymbol{x}(t) \in \mathcal{X}^*$.

**Proof sketch of Corollary 1.** Proposition 1 shows that when $r$ is fixed, for $\forall i \neq \sigma_{k_t}^t$, the value of $x_i(t)$ only depends on whether $\hat{\mu}_i(t) < \hat{\mu}_{\sigma_{k_t}^t}(t)$ or not. Similarly, for $\forall i \neq \sigma_k$, the value of $x_i^*$ only depends on whether $\mu_i < \mu_{\sigma_k}$ or not. This implies that, if $k_t = k, \sigma_{k_t}^t = \sigma_k$, and $\hat{\mu}_i(t) - \hat{\mu}_{\sigma_{k_t}^t}(t)$ have the same sign with $\mu_i - \mu_{\sigma_k}$, then $x_i(t) = x_i^*$. In other words, if $\mu_i > \mu_{\sigma_k} \Rightarrow \hat{\mu}_i(t) > \hat{\mu}_{\sigma_k}(t)$ and $\mu_i < \mu_{\sigma_k} \Rightarrow \hat{\mu}_i(t) < \hat{\mu}_{\sigma_k}(t)$ holds for $\forall i \in [N]$, then $\boldsymbol{x}(t) = \boldsymbol{x}^*$. When $||\hat{\boldsymbol{\mu}}(t) - \boldsymbol{\mu}||_\infty < \frac{\epsilon}{2}$, by the definition of $\epsilon$, for $\forall i$ we have $\mu_i > \mu_{\sigma_k} \Rightarrow \mu_i \geq \mu_{\sigma_k} + \epsilon \Rightarrow \hat{\mu}_i(t) > \mu_i - \frac{\epsilon}{2} \geq \mu_{\sigma_k} + \frac{\epsilon}{2} > \hat{\mu}_{\sigma_k}(t)$ and $\mu_i < \mu_{\sigma_k} \Rightarrow \mu_i \leq \mu_{\sigma_k} - \epsilon \Rightarrow \hat{\mu}_i(t) < \mu_i + \frac{\epsilon}{2} \leq \mu_{\sigma_k} - \frac{\epsilon}{2} < \hat{\mu}_{\sigma_k}(t)$. Then $\boldsymbol{x}(t) = \boldsymbol{x}^*$.

With Corollary 1, we derive another constant regret bound of UCB-LP in the following theorem.

**Theorem 3** *UCB-LP also guarantees the following regret bound for fairness constraints,*

$$Regret_T \leq \frac{64m}{r_{\min}^2 \epsilon^2} \ln^2 \left( \frac{64}{r_{\min}^2 \epsilon^2} \right) + \frac{mN\pi^2}{2}.$$

In the supplementary material, we show that $\Delta_{\min} = \epsilon = \Delta$ when our formulation reduces to the classical MAB setting, which suggests that $\Delta_{\min}$ and $\epsilon$ are both reasonable generalizations of $\Delta$. Although $\epsilon$ and $\Delta_{\min}$ are generally incomparable, we believe that the regret bound given by Theorem 3 is tighter than Theorem 2. To provide evidences for this, in supplementary material, we investigate the closed form expression of $\Delta_{\min}$. We show that $\epsilon = \Delta \geq \Delta_{\min} = \Delta(1 - \sum_i r_i)$ in the special case of $m = 1$. Intuitively, $\epsilon$ characterizes the structural properties of fairness constraints, while $\Delta_{\min}$ is more general and applies to all linear constraints.

**Remark 1** *LP-sensitivity arguments are not new in bandit analyses and was firstly shown in [42] which used a technique based on LP-sensitivity to analyze the UcbBwK algorithm [1] for BwK problem in the single-arm setting. Here we would like to point out that the technique based on LP-sensitivity we used is completely different from theirs. Specifically, their analysis relies on the assumptions of single resource and that the best distribution over arms reduces to the best fixed arm, while we do not require any assumptions. Moreover, our technique is also non-standard as our sensitivity analysis (Lemma 2, Corollary 1) takes full advantage of the fairness structure (e.g., closed-form solution of LP). Thus, our result is sharper than standard results about LP sensitivity and leads to the $O(1)$ regret bound.*

---

**Algorithm 1** UCB-PLLP

---

1: **Initialization:** $\mathcal{A} = \{\boldsymbol{x} | \boldsymbol{x} \in \{0, 1\}^N, ||\boldsymbol{x}||_1 \leq m\}$
2: **for** round $t = 1, ...T - 1$ **do**
3:     Compute UCBs: $\hat{\mu}_i(t) = \min\{\overline{\mu}_i(t) + \sqrt{\frac{2 \ln t}{h_i(t)}}, 1\}$, $\forall i$.
4:     Update the primal iterate: $\boldsymbol{a}(t) = \arg\max_{\boldsymbol{a} \in \mathcal{A}} \langle \hat{\boldsymbol{\mu}}(t) - \alpha_t \sum_{k=1}^{K} \nabla g_k(\boldsymbol{a}(t-1)) Q_k(t), \boldsymbol{a} \rangle$
5:     Play arm $i$ and receive $f_i(t)$ if $a_i(t) = 1$.
6:     Update the virtual queues: $\boldsymbol{Q}(t+1) = [\boldsymbol{Q}(t) + \boldsymbol{g}(\boldsymbol{a}(t)) + \epsilon_t \boldsymbol{I}]^+$.
7:     Update the statistics: $h_i(t+1)$, $\overline{\mu}_i(t+1)$, $\forall i$.
8: **end for**

---

*Except [42] studies the BwK problem based on the LP methodology, [5] also develops an algorithm based on LP methodology to track the (contextual) blocking bandit problem wherein once an arm is pulled it cannot be played again for a fixed number of consecutive rounds. Here we argue that their LP-sensitivity-based proof techniques cannot obtain our regret bounds. Specifically, [5] utilizes techniques in [10] and [44], where [10] proposes the general CMAB framework and [44] improves upon its regret bounds. The core of their techniques is to maintain a set of counters for every arm $i$ and allocate the regret to arms according to these counters. However, this regret allocation leads to the arms in the support of LP solution associated with the same term while their probability of being triggered by the LP-solution is different. This coarseness makes the final regret bound to scale with the size of the support, transforming the regret bound from our result of $\Theta(mN \ln T / \Delta_{\min})$ to $\Theta(N^2 \ln T / \Delta_{\min})$. In contrast, our analysis is more fine-grained since we do not maintain a counter for every arm $i$ and allocate the regret according to the counter value, but allocate the regret to arm $i$ according to the number of times it is pulled directly. More specifically, we only allocate the regret to arms in $V(t) := \{i : h_i(t) \leq \frac{32m^2 \ln t}{\Delta_{\min}^2}\}$, i.e., the "dominant arms" whose pulls is no more than $O(m^2 \ln t / \Delta_{\min}^2)$ times. Thus, each arm is bounded according to its probability of being triggered, and one no longer needs to distinguish arms in the support and other arms.*

**Remark 2** *When the long-term constraints only need to be satisfied in expectation, someone may adopt the well-established linear bandit algorithms like LinUCB [14] to track and identify the optimal sampling distribution over arms (optimal extreme point). We pointed out that applying LinUCB to our model involves solving an NP-hard optimization problem and fails to achieve a logarithmic regret bound of $O(mN \log T)$ as our algorithm UCB-LP does. In contrast, UCB-LP has a much lower computational complexity than the reduced LinUCB algorithm since it is polynomial-time computable. Furthermore, UCB-LP can guarantee a bounded regret for fairness setting. The reason why LinUCB produces sub-optimal results is that LinUCB ignores the structural properties of the constraints (e.g., concentration property bonus caused by the fairness constraints) and observations about each individual pulled arm.*

## 2.4 $\tilde{O}(N)$ **running time version of UCB-LP and its theoretical performance**

As mentioned earlier, UCB-LP might be computationally inefficient when constraint function $\boldsymbol{g}(\cdot)$ is complicated. In this section, we present the low computational-complexity version of UCB-LP, the UCB-PLLP algorithm, for problems of combinatorial bandits with linear long-term constraint. Note that the main computational bottleneck of UCB-LP is caused by the constraint $\boldsymbol{g}(\boldsymbol{x}) \leq \boldsymbol{0}$. To be computationally efficient, we optimize the (partial) Lagrangian of LP (4) at each round $t$:

$$\max_{\boldsymbol{x} \in R^N} \mathcal{L}_t(\boldsymbol{x}, \boldsymbol{\lambda}_t) = \langle \hat{\boldsymbol{\mu}}(t), \boldsymbol{x} \rangle - \boldsymbol{\lambda}_t^T \boldsymbol{g}(\boldsymbol{x}) \quad s.t. \quad \boldsymbol{0} \leq \boldsymbol{x} \leq \boldsymbol{1}, \; ||\boldsymbol{x}||_1 \leq m. \tag{8}$$

In (8) $\boldsymbol{\lambda}_t$ is the Lagrange multiplier associated with the constraints at round $t$. The main challenge is how to design $\boldsymbol{\lambda}_t$ to make a good balance between reward maximization and long-term constraints satisfaction. To address this challenge, we construct a virtual queue $\boldsymbol{Q}(t)$ to keep track of the "debt" of constraint violations up to round $t$, i.e., $\boldsymbol{Q}(t) = [\boldsymbol{Q}(t-1) + \boldsymbol{g}(\boldsymbol{a}(t-1))]^+$, and let $\boldsymbol{\lambda}_t = \alpha_t \boldsymbol{Q}(t)$, where $\alpha_t$ could be time-varying. However, this Lagrange multiplier deign may lead to large constraint violations as $\boldsymbol{g}(\cdot)$ becomes a "soft" constraint in LP (8) while in LP (4) it is a "hard" constraint.

To yield lower constraint violations, we incorporate the virtual queue update with "pessimistic" mechanism [35] so that the virtual queues overestimate the constraint violations, which is a novelty of our UCB-PLLP. Although the idea of using a "pessimistic" mechanism has been exploited in linear

(contextual) bandits with safety constraints (e.g., [2, 38], etc), the pessimism in our algorithm is achieved via adding a time-varying tightness constant to virtual queue update, i.e., $\boldsymbol{Q}(t) = [\boldsymbol{Q}(t-1) + \boldsymbol{g}(\boldsymbol{a}(t-1)) + \epsilon_{t-1}\boldsymbol{I}]^+$, which is completely different from the pessimism in previous works. Since $\boldsymbol{g}(\cdot)$ is linear, the optimization problem (8) has an integral closed-form solution and is equivalent to the following optimization problem:

$$\max_{\boldsymbol{x}} \ \langle \hat{\boldsymbol{\mu}}(t) - \alpha_t \sum_{k\in[K]} \nabla g_k(\boldsymbol{a}(t-1))Q_k(t), \boldsymbol{x}\rangle \quad s.t. \quad \boldsymbol{x} \in [0,1]^N, \ ||\boldsymbol{x}||_1 \le m.$$

$$\Longleftrightarrow \max_{\boldsymbol{a}} \ \langle \hat{\boldsymbol{\mu}}(t) - \alpha_t \sum_{k\in[K]} \nabla g_k(\boldsymbol{a}(t-1))Q_k(t), \boldsymbol{a}\rangle \quad s.t. \quad \boldsymbol{a} \in \{0,1\}^N, \ ||\boldsymbol{a}||_1 \le m. \tag{9}$$

We illustrate this algorithmic approach in Algorithm 1. Note that if we set $\alpha_t = \eta$ and $\epsilon_t = 0$, the algorithm LBF in [29] is our special case as $\nabla g_k(\cdot) = -\boldsymbol{e}_k$ and $K = N$ for fairness constraints setting. Apparently, the computational-complexity of UCB-PLLP is essentially the same as choosing the top $m$ arms with maximum positive compound value, which is $\tilde{O}(N)$.

**Theoretical performance of UCB-PLLP.** Here we present the regret bound and constraint violations for UCB-PLLP. Our result relies on a mild assumption of Slater condition (Interior condition), i.e., there exists a $\delta > 0$ and $\hat{\boldsymbol{x}} \in \mathcal{D}$ such that $\boldsymbol{g}(\hat{\boldsymbol{x}}) \le -\delta\boldsymbol{I}$. Note that Slater condition automatically holds for fairness constraints as $r_i < 1, \ \forall i$, and $\sum_{i\in[N]} r_i < m$. It is also a default assumption in BwK literature (They all assume the null arm denoted by 0 exists, i.e, $\lambda_{i,0} = 0, \ \forall i \in [K]$).

**Theorem 4** *Set $\epsilon_t = O(\frac{\delta}{\sqrt{t}})$ and $\alpha_t = O(\frac{N}{\delta\sqrt{t}})$, then UCB-PLLP achieves*

$$Regret_T \le \tilde{O}(m\sqrt{T}), \quad \Pr[\text{Vio}_k(\tau) > 0] \le O(e^{-\delta\sqrt{\tau}}), \ \forall k \in [K], \ \tau \le T.$$

We give the proof of Theorem 4 in supplementary material. Our proof technique is based on the Lyapunov-drift analysis for queueing systems. The bounds in Theorem 4 are sharp in the perspective that the regret bound matches the problem-independent regret of UCB1 algorithm in standard MAB setting, and the probability of constraints not being violated converges to 1, i.e, holds asymptotically almost surely.

**Intuition behind Theorem 4.** Here we explain the intuition why UCB-PLLP has such performance guarantees. Since $\alpha = O(1/\sqrt{t})$, the reward term dominates the whole term in (9) when $Q_k(t) = o(\sqrt{t})$. If $Q_k(t) = \omega(\sqrt{t})$, the term containing virtual queues dominates the reward term, and UCB-PLLP tends to reduce the virtual queues $Q_k(t)$. Slater's condition implies that there exists a policy that can reduce $Q_k(t)$ by a constant (related to $\delta$) in each round. Therefore, the algorithm takes at most $O(\sqrt{t})$ rounds to reduce $Q_k(t)$ to $o(\sqrt{t})$, which may lead to $O(\sqrt{t})$ increase of the regret. Thus, we could derive that $Q_k(t) = O(\sqrt{t})$. Recall that $Q_k(t+1) \ge \sum_{t=1}^T g_k(\boldsymbol{a}(t)) + \sum_{t=1}^T \epsilon_t \Rightarrow \text{Vio}_k(T) = \sum_{t=1}^T g_k(\boldsymbol{a}(t)) \le Q_k(T+1) - \sum_{t=1}^T \epsilon_t$, we can obtain zero constraint violation via choosing proper $\epsilon_t$. Then, the high probability constraint violation guarantee is established by bounding the exponential moment of the virtual queues since $\Pr(\text{Vio}_k(T) \ge 0) \le \Pr(Q_k(T+1) \ge \sum_{t=1}^T \epsilon_t) \le E[e^{||\boldsymbol{Q}(T+1)||_1}]/e^{\sum_{t=1}^T \epsilon_t}$.

**Remark 3** *Although the virtual queue techniques have been used for various constrained online learning problems in the literature (e.g., [8, 9, 29, 43, 20, 49, 27, 35, 32, 31, 34]), our virtual queue update rule differs from theirs in the pessimistic mechanism via adding a time-varying tightness constant. Beyond the update rule of virtual queue, we also employ some new techniques in our Lyapunov analysis. For example, we establish a bound on the exponential moment of the virtual queue length (Lemma 6), which is the central focus of our high-probability guarantee on zero constraint violations. We also establish an upper bound on the $\epsilon_t$-tight term (incurred by our pessimistic mechanism) via comparing the optimal solution to the original LP problem and that to its -tightened version based on LP-sensitivity (lemmas 3 and 4). These analysis techniques are not present in these works.*

## 2.5 Experiments

The results of our numerical experiments are given in the supplementary material.

## Acknowledgments and Disclosure of Funding

The work of Siwei Wang is supported in part by the National Natural Science Foundation of China Grant 62106122.

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
