# Combinatorial Bandits with Linear Constraints: Beyond Knapsacks and Fairness

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

} := \{x | x \in [0,1]^N, ||x||_1 \leq m, g(x) \leq 0\}$. In this paper, for any vector $v$, we use $v_i$ to denote its $i^{\text{th}}$ coordinate. For any event $E$, we use $\overline{E}$ to denote its negation.

**Extreme points and general sub-optimality measure.** Note that the feasible region $\mathcal{D}$ to LP (2) is a convex polytope, and we let $\mathcal{B}$ be the set of its extreme points. An extreme point of $\mathcal{D}$ is a point in $\mathcal{D}$ which does not lie in any open line segment joining two points of $\mathcal{D}$. It is well-known in the theory of LP that extreme points and basic feasible solutions are equivalent, and any LP attains its optimal value at an extreme point [6]. Recall that $x^*$ is an optimal solution to LP (2), and we define the sub-optimality gap for any $x \in \mathcal{B}$ as $\Delta_x := \langle \mu, x^* \rangle - \langle \mu, x \rangle$. Define $\Delta_{\min} := \min_{

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

# A Proofs for Subsection 2.2

## A.1 $O(N \log N)$ running time method for decomposition

Recall that $\mathcal{A} = \{\boldsymbol{a} \mid \boldsymbol{a} \in \{0,1\}^N, ||\boldsymbol{a}||_1 \leq m\}$ and we define the $m$-set as $\tilde{\mathcal{A}} = \{\boldsymbol{a} \mid \boldsymbol{a} \in \{0,1\}^N, ||\boldsymbol{a}||_1 = m\}$. Note that $\tilde{\mathcal{A}} \subseteq \mathcal{A}$.

[53] showed in their Appendix B.2 that, for any $\boldsymbol{x} \in \text{Conv}(\tilde{\mathcal{A}})$, one can compute a distribution $\pi$ over set $\tilde{\mathcal{A}}$ such that $\sum_{\boldsymbol{a} \in \tilde{\mathcal{A}}} \pi(\boldsymbol{a}) \cdot \boldsymbol{a} = \boldsymbol{x}$, and sample the action vector $\boldsymbol{a} \sim \pi$ in $O(N \log N)$ time. In our setting, we only need to do the same thing for $\mathcal{A}$. Now we show that their sampling scheme can be easily generalized from $\tilde{\mathcal{A}}$ to $\mathcal{A}$.

In fact, to sample an action vector with an expected value of $\boldsymbol{x}$, if $\boldsymbol{x} = \boldsymbol{0}$, then the sampling is trivial, otherwise we first compute a vector $\tilde{\boldsymbol{x}} := \boldsymbol{x} \cdot \frac{m}{||\boldsymbol{x}||_1}$. Note that $\tilde{\boldsymbol{x}} \in \text{Conv}(\tilde{\mathcal{A}})$, so we can call the sampling scheme in [53] with input $\tilde{\boldsymbol{x}}$ as a subroutine. Denote the output of their sampling scheme (i.e., the sampled action vector) as $\tilde{\boldsymbol{a}}$. Then $\tilde{\boldsymbol{a}} \in \tilde{\mathcal{A}} \subseteq \mathcal{A}$.

Finally, with probability $1 - \frac{||\boldsymbol{x}||_1}{m}$, we choose action vector $\boldsymbol{a} = \boldsymbol{0}$ as the final outcome of sampling; and with probability $\frac{||\boldsymbol{x}||_1}{m}$, we choose action vector $\tilde{\boldsymbol{a}}$ as the final outcome of sampling. It can be easily verified that the expected value of the sampled action vector equals to $\boldsymbol{x}$.

## A.2 Proof of Theorem 1

*Proof:* Note that (6) is straightforward from LP (4) and the fact that $\boldsymbol{g}(\cdot)$ is linear. Now we are going to proving (5). Since $E_{\pi_t}[a_i(t)] = x_i(t)$ and the two random variables $f_i(t)$ and $a_i(t)$ are independent, therefore,

$$
\begin{aligned}
E[R_t] &= \sum_{i \in [N]} E[f_i(t) a_i(t)] = \sum_{i \in [N]} E[f_i(t)] E[a_i(t)] \\
&= \sum_{i \in [N]} \mu_i(t) E[x_i(t)] = E[\langle \boldsymbol{\mu}, \boldsymbol{x}(t) \rangle].
\end{aligned}
\tag{10}
$$

For convenience, we redefine $h_i(t) := \sum_{\tau=1}^{t-1} a_i(\tau) + 1$, i.e., $h_i(t) \geq 1$, and related confidence bounds at time $t$ still hold with probability $1 - t^2$. We also define $\boldsymbol{w}(t) = (\sqrt{\frac{2 \ln t}{h_i(t)}}, .., \sqrt{\frac{2 \ln t}{h_N(t)}})^\top$. When $\boldsymbol{x} \notin \mathcal{X}^*$, the following inequalities hold with high-probability $1 - t^2$:

$$
\langle \boldsymbol{\mu}, \boldsymbol{x}^* \rangle \leq \langle \hat{\boldsymbol{\mu}}(t), \boldsymbol{x}(t) \rangle,
\tag{11}
$$

$$
\langle \hat{\boldsymbol{\mu}}(t), \boldsymbol{x}(t) \rangle \geq \langle \boldsymbol{\mu}(t), \boldsymbol{x}(t) \rangle + \Delta_{\boldsymbol{x}(t)},
\tag{12}
$$

$$
\langle \boldsymbol{w}, \boldsymbol{x}(t) \rangle = \sum_{i \in [N]} x_i(t) \sqrt{\frac{2 \ln t}{h_i(t)}} \geq \Delta_{\boldsymbol{x}(t)}/2.
\tag{13}
$$

Where (11) and (12) are due to

$$
\langle \hat{\boldsymbol{\mu}}(t), \boldsymbol{x}(t) \rangle \overset{\text{(LP property)}}{\geq} \langle \hat{\boldsymbol{\mu}}(t), \boldsymbol{x}^* \rangle
$$
$$
\overset{\text{(high-prob)}}{\geq} \langle \boldsymbol{\mu}, \boldsymbol{x}^* \rangle \geq \langle \boldsymbol{\mu}, \boldsymbol{x}(t) \rangle + \Delta_{\boldsymbol{x}(t)};
\tag{14}
$$

(13) is given by continuing from (14), i.e.,

$$
\begin{aligned}
&\langle \hat{\boldsymbol{\mu}}(t), \boldsymbol{x}(t) \rangle \geq \langle \boldsymbol{\mu}, \boldsymbol{x}(t) \rangle + \Delta_{\boldsymbol{x}(t)} \\
&\Rightarrow \langle \hat{\boldsymbol{\mu}}(t) - \boldsymbol{\mu}, \boldsymbol{x}(t) \rangle \geq \Delta_{\boldsymbol{x}(t)} \\
&\Rightarrow \langle 2\boldsymbol{w}, \boldsymbol{x}(t) \rangle \geq \Delta_{\boldsymbol{x}(t)} \\
&\Rightarrow \langle \boldsymbol{w}, \boldsymbol{x}(t) \rangle \geq \Delta_{\boldsymbol{x}(t)}/2.
\end{aligned}
$$

We first decompose the regret as:

$$Regret_T \leq E[\sum_{t=1}^{T}(\langle \boldsymbol{\mu}, \boldsymbol{x}^* \rangle - \langle \boldsymbol{\mu}, \boldsymbol{x}(t) \rangle)] = E[\sum_{t=1}^{T}(\langle \boldsymbol{\mu}, \boldsymbol{x}^* \rangle - \langle \boldsymbol{\mu}, \boldsymbol{x}(t) \rangle)\,\mathbf{I}\{\boldsymbol{x}(t) \notin \mathcal{X}^*\}]$$

$$\leq E[\sum_{t=1}^{T}\langle \hat{\boldsymbol{\mu}}(t) - \boldsymbol{\mu}, \boldsymbol{x}(t)\rangle\mathbf{I}\{\boldsymbol{x}(t) \notin \mathcal{X}^*\}] + E[\sum_{t=1}^{T}(\langle \boldsymbol{\mu}, \boldsymbol{x}^* \rangle - \langle \hat{\boldsymbol{\mu}}(t), \boldsymbol{x}(t)\rangle)\,\mathbf{I}\{\boldsymbol{x}(t) \notin \mathcal{X}^*\}]$$

$$\overset{(11)}{\leq} E[\sum_{t=1}^{T}\langle \hat{\boldsymbol{\mu}}(t) - \boldsymbol{\mu}, \boldsymbol{x}(t)\rangle\mathbf{I}\{\boldsymbol{x}(t) \notin \mathcal{X}^*\}] + m\sum_{t=1}^{T}t^{-2}$$

$$\overset{(\text{Hoeffding inequality})}{\leq} E[\sum_{t=1}^{T}\langle 2\boldsymbol{w}, \boldsymbol{x}(t)\rangle\mathbf{I}\{\boldsymbol{x}(t) \notin \mathcal{X}^*\}] + m\sum_{t=1}^{T}t^{-2} + m\sum_{t=1}^{T}t^{-2}$$

$$\overset{(11)}{\leq} 2E[\sum_{t=1}^{T}\langle \boldsymbol{w}, \boldsymbol{x}(t)\rangle\mathbf{I}\{\boldsymbol{x}(t) \notin \mathcal{X}^*, \langle \boldsymbol{w}, \boldsymbol{x}(t)\rangle \geq \Delta_{\boldsymbol{x}(t)}/2\}]$$

$$+ 2m\sqrt{2\ln T}\sum_{t=1}^{T}\Pr\{\langle \boldsymbol{w}, \boldsymbol{x}(t)\rangle < \Delta_{\boldsymbol{x}(t)}/2 \mid \boldsymbol{x}(t) \notin \mathcal{X}^*\} + 2m\sum_{t=1}^{T}t^{-2}$$

$$\overset{(13)}{\leq} 2E[\sum_{t=1}^{T}\langle \boldsymbol{w}, \boldsymbol{x}(t)\rangle\mathbf{I}\{\boldsymbol{x}(t) \notin \mathcal{X}^*, \langle \boldsymbol{r}, \boldsymbol{x}(t)\rangle \geq \Delta_{\boldsymbol{x}(t)}/2\}] + 2m\sqrt{2\ln T}\sum_{t=1}^{T}t^{-2} + 2m\sum_{t=1}^{T}t^{-2}$$

$$\leq 2E[\sum_{t=1}^{T}\langle \boldsymbol{w}, \boldsymbol{x}(t)\rangle\mathbf{I}\{\boldsymbol{x}(t) \notin \mathcal{X}^*, \langle \boldsymbol{r}, \boldsymbol{x}(t)\rangle \geq \Delta_{\boldsymbol{x}(t)}/2\}] + 2m(\sqrt{2\ln T}+1)\sum_{t=1}^{T}t^{-2}$$

$$= 2E[\sum_{t=1}^{T}\sum_{i\in[N]}x_i(t)\sqrt{\frac{2\ln t}{h_i(t)}}\mathbf{I}\{\boldsymbol{x}(t) \notin \mathcal{X}^*, \langle \boldsymbol{r}, \boldsymbol{x}(t)\rangle \geq \Delta_{\boldsymbol{x}(t)}/2\}] + 2m(\sqrt{2\ln T}+1)\sum_{t=1}^{T}t^{-2}$$

Define $V(t) = \{i : h_i(t) \leq \frac{32m^2\ln t}{\Delta_{\min}^2}\}$. When $\mathbf{I}\{\boldsymbol{x}(t) \notin \mathcal{X}^*, \langle \boldsymbol{r}, \boldsymbol{x}(t)\rangle \geq \Delta_{\boldsymbol{x}(t)}/2\} = 1$ we have

$$\sum_{i\in[N]}x_i(t)\sqrt{\frac{2\ln t}{h_i(t)}} \leq 2\sum_{i\in V(t)}x_i(t)\sqrt{\frac{2\ln t}{h_i(t)}}, \tag{15}$$

due to the facts that

$$\sum_{i\in[N]/V(t)}x_i(t)\sqrt{\frac{2\ln t}{h_i(t)}}$$

$$\leq \sum_{i\in[N]/V(t)}x_i(t)\frac{\Delta_{\min}}{4m}$$

$$\leq \sum_{i\in[N]}x_i(t)\frac{\Delta_{\min}}{4m} \leq \frac{\Delta_{\min}}{4} \leq \frac{\Delta_{\boldsymbol{x}(t)}}{4},$$

and

$$\sum_{i\in[N]}x_i(t)\sqrt{\frac{2\ln t}{h_i(t)}} \geq \frac{\Delta_{\boldsymbol{x}(t)}}{2}.$$

Therefore,

$$Regret_T \leq 2E[\sum_{t=1}^{T}\sum_{i\in[N]}x_i(t)\sqrt{\frac{2\ln t}{h_i(t)}}\mathbf{I}\{\boldsymbol{x}(t) \notin \mathcal{X}^*, \langle \boldsymbol{r}, \boldsymbol{x}(t)\rangle \geq \Delta_{\boldsymbol{x}(t)}/2\}] + 2m(\sqrt{2\ln T}+1)\sum_{t=1}^{T}t^{-2}$$

$$\leq 4E[\sum_{t=1}^{T}\sum_{i\in V(t)}x_i(t)\sqrt{\frac{2\ln t}{h_i(t)}}\mathbf{I}\{\boldsymbol{x}(t) \notin \mathcal{X}^*, \langle \boldsymbol{r}, \boldsymbol{x}(t)\rangle \geq \Delta_{\boldsymbol{x}(t)}/2\}] + 2m(\sqrt{2\ln T}+1)\sum_{t=1}^{T}t^{-2}$$

$$\leq 4E[\sum_{t=1}^{T}\sum_{i\in V(t)}x_i(t)\sqrt{\frac{2\ln t}{h_i(t)}}] + 2m(\sqrt{2\ln T}+1)\sum_{t=1}^{T}t^{-2}.$$

$$\tag{16}$$

Define $G(i) = \{t : i \in V(t)\}$, and $T_i = \arg\max_{\tau \in V(t)} \tau$. Then we obtain

$$Regret_T \leq 4E[\sum_{t=1}^{T} \sum_{i \in V(t)} x_i(t)\sqrt{\frac{2\ln t}{h_i(t)}}] + 2m(\sqrt{2\ln T} + 1)\sum_{t=1}^{T} t^{-2}$$

$$\leq 4E[\sum_{i \in N} \sum_{t \in G(i)} x_i(t)\sqrt{\frac{2\ln t}{h_i(t)}}] + 2m(\sqrt{2\ln T} + 1)\sum_{t=1}^{T} t^{-2}$$

$$\leq 4E[\sum_{i \in N} \sum_{t=1}^{T_i} x_i(t)\sqrt{\frac{2\ln t}{h_i(t)}}] + 2m(\sqrt{2\ln T} + 1)\sum_{t=1}^{T} t^{-2}$$

$$\overset{\text{lemma 1}}{\leq} 4 \cdot 3 \sum_{i \in [N]} \sqrt{2\ln T_i(h_i(T_i) + 1)} + 2m(\sqrt{2\ln T} + 1)\sum_{t=1}^{T} t^{-2} \qquad (17)$$

$$\overset{T_i \in V(T_i)}{\leq} 4 \cdot 3 \sum_{i \in [N]} \sqrt{2\ln T_i(\frac{32m^2 \ln T_i}{\Delta_{\min}^2} + 1)} + 2m(\sqrt{2\ln T} + 1)\sum_{t=1}^{T} t^{-2}$$

$$\leq 4 \cdot 3 \sum_{i \in [N]} \sqrt{2\ln T(\frac{32m^2 \ln T}{\Delta_{\min}^2} + 1)} + 2m(\sqrt{2\ln T} + 1)\sum_{t=1}^{T} t^{-2}$$

$$\leq 12 \cdot 8 \frac{mN}{\Delta_{\min}} \ln T + 12N \cdot \sqrt{2\ln T} + \frac{m\pi^2}{3}m(\sqrt{2\ln T} + 1).$$

# B   Proofs for Subsection 2.3

## B.1   Proof of Theorem 2

*Proof:* Firstly, plug (10) into the definition of regret in (3), we decompose the regret as

$$Regret_T \leq E\left[\sum_{t=1}^{T} (\langle \boldsymbol{\mu}, \boldsymbol{x}^* \rangle - \langle \boldsymbol{\mu}, \boldsymbol{x}(t) \rangle)\right] \leq m\sum_{t=1}^{T} \Pr[\boldsymbol{x}(t) \notin \mathcal{X}^*]$$

$$\leq mc + m\sum_{t=c+1}^{T} \Pr[\boldsymbol{x}(t) \notin \mathcal{X}^*]$$

$$\overset{(a)}{\leq} mc + m\sum_{t=c+1}^{T} \Pr[||\hat{\boldsymbol{\mu}}(t) - \boldsymbol{\mu}||_1 \geq \Delta_{\min}] \qquad (18)$$

$$\overset{(b)}{\leq} mc + m\sum_{i=1}^{N} \sum_{t=c+1}^{T} \Pr[|\hat{\mu}_i(t) - \mu_i| \geq \Delta_{\min}/N].$$

In (18), $c$ is a parameter to be determined later, (a) holds due to Lemma 2 and (b) is because

$$||\hat{\boldsymbol{\mu}}(t) - \boldsymbol{\mu}||_1 \geq \Delta_{\min} \Rightarrow \max_{i \in [N]} |\hat{\mu}_i(t) - \mu_i| \geq \Delta_{\min}/N.$$

Note that $\hat{\mu}_i(t) - \mu_i \geq \Delta_{\min}/N$ implies that at least one of the following two events must happen:

$$F_i(t) = \left\{\overline{\mu}_i(t) \geq \mu_i + \sqrt{\frac{2\ln t}{h_i(t)}}\right\}, \quad J_i(t) := \left\{\sqrt{\frac{2\ln t}{h_i(t)}} \geq \frac{\Delta_{\min}}{2N}\right\}.$$

The probability of the first event could be bounded with Chernoff-Hoeffding inequality: $\Pr[F_i(t)] \leq t^{-2}$. For the second event $J_i(t)$, notice that

$$\sqrt{\frac{2\ln t}{h_i(t)}} \geq \frac{\Delta_{\min}}{2N} \iff h_i(t) \leq \frac{8N^2}{\Delta_{\min}^2} \ln t. \qquad (19)$$

The next step is to set $c$ to be sufficiently large such that

$$\frac{8N^2}{\Delta_{\min}^2} \ln t + \sqrt{t\ln t} \leq r_i t, \forall t \geq c. \qquad (20)$$

We set $c = \frac{32N^2}{r_{\min}^2 \Delta_{\min}^2} \ln^2\left(\frac{32N^2}{r_{\min}^2 \Delta_{\min}^2}\right)$ and claim that it satisfies (20). The proof is by basic algebra and we defer the details to Appendix B.3.

According to (19) and (20), $\forall t \geq c$, $J_i(t)$ implies

$$\sum_{\tau=1}^{t} a_i(\tau) = h_i(t) \leq r_i t - \sqrt{t \ln t} \overset{(a)}{\leq} \sum_{\tau=1}^{t} x_i(\tau) - \sqrt{t \ln t}, \tag{21}$$

where (a) in (21) is because the fairness constraints guarantee that $x_i(\tau) \geq r_i, \forall \tau \in [t]$. Consequently,

$$\forall t \geq c, J_i(t) \Rightarrow \sum_{\tau=1}^{t}(a_i(\tau) - x_i(\tau)) \leq -\sqrt{t \ln t}. \tag{22}$$

Define a filtration up to time $t$: $\mathcal{H}_t = \{(\boldsymbol{f}(\tau), \boldsymbol{a}(\tau))\}_{\tau=1}^{t}$, where $\boldsymbol{f}(\tau) := (f_1(\tau), f_2(\tau), \ldots, f_N(\tau))^\top$. Note that $a_i(t) - x_i(t)$ is a martingale difference with respect to the filtration $\mathcal{H}_t$. From Azuma-Hoeffding inequality we have, $\forall t \geq c$,

$$\Pr[J_i(t)] \overset{(22)}{\leq} \Pr\left[\sum_{\tau=1}^{t}(a_i(\tau) - x_i(\tau)) \leq -\sqrt{t \ln t}\right] \leq t^{-2}.$$

Therefore, $\forall t \geq c$

$$\Pr[\hat{\mu}_i(t) - \mu_i \geq \Delta_{\min}/N] \leq \Pr[F_i(t)] + \Pr[J_i(t)] \leq \frac{2}{t^2}.$$

On the other hand, using Chernoff-Hoeffding inequality we get

$$\Pr[\hat{\mu}_i(t) - \mu_i \leq -\Delta_{\min}/N] \leq \Pr[\hat{\mu}_i(t) < \mu_i] \leq \frac{1}{t^2}.$$

Then $\Pr[|\hat{\mu}_i(t) - \mu_i| \geq \Delta_{\min}/N] \leq 3/t^2, \forall t \geq c$.

Continuing from (18), the regret can be bounded as

$$Regret_T \leq mc + m\sum_{i=1}^{N}\sum_{t=c+1}^{T} \Pr[|\hat{\mu}_i(t) - \mu_i| \geq \Delta_{\min}/N]$$

$$\leq mc + m\sum_{i=1}^{N}\sum_{t=c+1}^{T} \frac{3}{t^2} \leq mc + \frac{mN\pi^2}{2}$$

$$= \frac{32mN^2}{r_{\min}^2 \Delta_{\min}^2} \ln^2\left(\frac{32N^2}{r_{\min}^2 \Delta_{\min}^2}\right) + \frac{mN\pi^2}{2}.$$

This completes the proof.

## B.2 Proof of Lemma 2

*Proof:* To derive a contradiction, we assume that $\boldsymbol{x}(t) \notin \mathcal{X}^*$ when $\|\hat{\boldsymbol{\mu}}(t) - \boldsymbol{\mu}\|_1 < \Delta_{\min}$.

Note that $\boldsymbol{x}(t) \in (\mathcal{B} \setminus \mathcal{X}^*)$, then the definition of $\Delta_{\min}$ and $\boldsymbol{x}(t)$ implies

$$\boldsymbol{\mu} \cdot \boldsymbol{x}^* \geq \boldsymbol{\mu} \cdot \boldsymbol{x}(t) + \Delta_{\min}, \text{ and } \hat{\boldsymbol{\mu}}(t) \cdot \boldsymbol{x}(t) \geq \hat{\boldsymbol{\mu}}(t) \cdot \boldsymbol{x}^*.$$

Combining the above two inequality gives

$$(\hat{\boldsymbol{\mu}}(t) - \boldsymbol{\mu}) \cdot \boldsymbol{x}(t) \geq (\hat{\boldsymbol{\mu}}(t) - \boldsymbol{\mu}) \cdot \boldsymbol{x}^* + \Delta_{\min}$$
$$\Rightarrow (\hat{\boldsymbol{\mu}}(t) - \boldsymbol{\mu}) \cdot (\boldsymbol{x}(t) - \boldsymbol{x}^*) \geq \Delta_{\min},$$

which contradicts the conditions $\|\hat{\boldsymbol{\mu}}(t) - \boldsymbol{\mu}\|_1 < \Delta_{\min}$ and $\|\boldsymbol{x}(t) - \boldsymbol{x}^*\|_\infty \leq 1$.

## B.3 Proof of of the adequacy of $c$

Now we prove that, in the proof of Theorem 2, the constant $c$ defined as $c = \frac{32N^2}{r_{\min}^2 \Delta_{\min}^2} \ln^2\left(\frac{32N^2}{r_{\min}^2 \Delta_{\min}^2}\right)$ satisfies (20). For the simplicity of notation, define $\gamma = \frac{32N^2}{r_{\min}^2 \Delta_{\min}^2}$, then $c = \gamma \ln^2 \gamma$. First we show that $\frac{c}{\ln c} \geq \gamma$. In fact,

$$\frac{c}{\ln c} \geq \gamma \iff \frac{\gamma \ln^2 \gamma}{\ln \gamma + 2\ln\ln\gamma} \geq \gamma \iff \ln^2 \gamma \geq \ln \gamma + 2\ln\ln\gamma \iff \ln\gamma(\ln\gamma - 1) \geq 2\ln\ln\gamma. \tag{23}$$

The last inequality in (23) holds because $\ln \gamma \geq \ln(32) > 2$ and $\ln \gamma - 1 \geq \ln \ln \gamma$. Thus, $\frac{c}{\ln c} \geq \gamma$.

Since $t \geq c \geq 32 > e$ and the function $\frac{t}{\ln t}$ monotonically increases for $t > e$, we have $\frac{t}{\ln t} \geq \frac{c}{\ln c} \geq \gamma, \forall t \geq c$.

Define a shorthand $\omega := \sqrt{\frac{t}{\ln t}}$. Then $\omega = \sqrt{\frac{t}{\ln t}} \geq \sqrt{\gamma} = \frac{4\sqrt{2}N}{r_{\min}\Delta_{\min}}$. Consider $\forall t \geq c$, we have

$$(20): \frac{8N^2}{\Delta_{\min}^2}\ln t + \sqrt{t \ln t} \leq r_i t \iff \frac{8N^2}{\Delta_{\min}^2} + \sqrt{\frac{t}{\ln t}} \leq r_i \cdot \frac{t}{\ln t} \iff \frac{8N^2}{\Delta_{\min}^2} \leq (r_i\omega - 1)\omega \quad (24)$$

The definition of $r_{\min}$ implies that $r_i\omega - 1 \geq \frac{4\sqrt{2}Nr_i}{r_{\min}\Delta_{\min}} - 1 \geq \frac{4\sqrt{2}N}{\Delta_{\min}} - 1 \geq \frac{\sqrt{2}N}{\Delta_{\min}}$. Then $(r_i\omega - 1)\omega \geq \frac{\sqrt{2}N}{\Delta_{\min}} \cdot \frac{4\sqrt{2}N}{\Delta_{\min}} = \frac{8N^2}{\Delta_{\min}^2}$, the correctness of the last inequality in (24) has been verified. So (20) is proved.

### B.4 Proof of Proposition 1

*Proof:* Due to the definition of $k_t$, we have

$$\sum_{i<k_t}(1 - r_{\sigma_i^t}) < m - \sum_{i=1}^N r_i \Rightarrow x_{\sigma_{k_t}^t}(t) = m + 1 - k_t - \sum_{i=k_t+1}^N r_{\sigma_i} > r_{\sigma_{k_t}},$$

$$\text{and } \sum_{i=1}^{k_t}(1 - r_{\sigma_i^t}) \geq m - \sum_{i=1}^N r_i \Rightarrow x_{\sigma_{k_t}^t}(t) = m + 1 - k_t - \sum_{i=k_t+1}^N r_{\sigma_i} \leq 1.$$

The above two inequalities imply that $\boldsymbol{x}(t)$ is feasible, i.e., $\boldsymbol{x}(t) \in \mathcal{D}$. Then $\boldsymbol{x}(t)$ is an extreme point in $\mathcal{B}$ since there are $N$ constraints in LP (4) that are active (tight) at point $\boldsymbol{x}(t)$.

Now to prove this proposition, we only need to prove that for any $\boldsymbol{x}' \in \mathcal{D}$, $\langle \hat{\boldsymbol{\mu}}(t), \boldsymbol{x}(t) \rangle \geq \langle \hat{\boldsymbol{\mu}}(t), \boldsymbol{x}' \rangle$. In fact,

$$\langle \hat{\boldsymbol{\mu}}(t), \boldsymbol{x}(t) \rangle - \langle \hat{\boldsymbol{\mu}}(t), \boldsymbol{x}' \rangle = \sum_{i<k_t} \hat{\mu}_{\sigma_i^t}(t)(x_{\sigma_i^t}(t) - x'_{\sigma_i^t}) + \hat{\mu}_{\sigma_{k_t}^t}(t)(x_{\sigma_{k_t}^t}(t) - x'_{\sigma_{k_t}^t}) - \sum_{i>k_t} \hat{\mu}_{\sigma_i^t}(t)(x'_{\sigma_i^t} - x_{\sigma_i^t}(t))$$

$$\geq \sum_{i<k_t} \hat{\mu}_{\sigma_{k_t}^t}(t)(x_{\sigma_i^t}(t) - x'_{\sigma_i^t}) + \hat{\mu}_{\sigma_{k_t}^t}(t)(x_{\sigma_{k_t}^t}(t) - x'_{\sigma_{k_t}^t}) - \sum_{i>k_t} \hat{\mu}_{\sigma_{k_t}^t}(t)(x'_{\sigma_i^t} - x_{\sigma_i^t}(t))$$

$$= \hat{\mu}_{\sigma_{k_t}^t}(t)\left(\sum_{i=1}^N x_{\sigma_i^t}(t) - \sum_{i=1}^N x'_{\sigma_i^t}\right) = 0,$$

where the first inequality holds because $x_{\sigma_i^t}(t) = 1 \geq x'_{\sigma_i^t}$ for all $i < k_t$, and $x_{\sigma_i^t}(t) = r_{\sigma_i^t} \leq x'_{\sigma_i^t}$ for all $i > k_t$. Thus, $\boldsymbol{x}(t)$ is optimal and we complete the proof.

### B.5 Proof of Corollary 1

*Proof:* According to Proposition 1, when $\boldsymbol{r}$ is fixed, for $\forall i \neq \sigma_{k_t}^t$, the value of $x_i(t)$ only depends on whether $\hat{\mu}_i(t) < \hat{\mu}_{\sigma_{k_t}^t}(t)$ or not. Similarly, for $\forall i \neq \sigma_k$, the value of $x_i^*$ only depends on whether $\mu_i < \mu_{\sigma_k}$ or not. This implies that, if $k_t = k, \sigma_{k_t}^t = \sigma_k$, and $\hat{\mu}_i(t) - \hat{\mu}_{\sigma_{k_t}^t}(t)$ have the same sign with $\mu_i - \mu_{\sigma_k}$, then $x_i(t) = x_i^*$. In other words, if

$$\mu_i > \mu_{\sigma_k} \Rightarrow \hat{\mu}_i(t) > \hat{\mu}_{\sigma_k}(t)$$

and

$$\mu_i < \mu_{\sigma_k} \Rightarrow \hat{\mu}_i(t) < \hat{\mu}_{\sigma_k}(t)$$

holds for $\forall i \in [N]$, then $\boldsymbol{x}(t) = \boldsymbol{x}^*$.

When $||\hat{\boldsymbol{\mu}}(t) - \boldsymbol{\mu}||_\infty < \frac{\epsilon}{2}$, by the definition of $\epsilon$, for $\forall i$ we have

$$\mu_i > \mu_{\sigma_k} \Rightarrow \mu_i \geq \mu_{\sigma_k} + \epsilon \Rightarrow \hat{\mu}_i(t) > \mu_i - \frac{\epsilon}{2} \geq \mu_{\sigma_k} + \frac{\epsilon}{2} > \hat{\mu}_{\sigma_k}(t)$$

and

$$\mu_i < \mu_{\sigma_k} \Rightarrow \mu_i \leq \mu_{\sigma_k} - \epsilon \Rightarrow \hat{\mu}_i(t) < \mu_i + \frac{\epsilon}{2} \leq \mu_{\sigma_k} - \frac{\epsilon}{2} < \hat{\mu}_{\sigma_k}(t).$$

Then $\boldsymbol{x}(t) = \boldsymbol{x}^*$.

## B.6 Proof of Theorem 3

*Proof:* In the same manner to (18), except that corollary 1 instead of Lemma 2 is applied, we have

$$
\begin{aligned}
Regret_T &\leq m \sum_{t=1}^{T} \Pr[\boldsymbol{x}(t) \notin \mathcal{X}^*] \\
&\leq mc' + m \sum_{t=c'+1}^{T} \Pr[\boldsymbol{x}(t) \notin \mathcal{X}^*] \\
&\leq mc' + m \sum_{t=c'+1}^{T} \Pr\left[||\hat{\boldsymbol{\mu}}(t) - \boldsymbol{\mu}||_\infty \geq \frac{\epsilon}{2}\right] \\
&\leq mc' + m \sum_{i=1}^{N} \sum_{t=c'+1}^{T} \Pr[|\hat{\mu}_i(t) - \mu_i| \geq \frac{\epsilon}{2}].
\end{aligned}
\tag{25}
$$

where $c'$ is still a parameter to be determined later, as in the the proof of Theorem 2.

Note that the last line in (25) is just the last line in (18), except that $\Delta_{\min}/N$ is replaced with $\frac{\epsilon}{2}$. Therefore we can set $c' = \frac{64}{r_{\min}^2 \epsilon^2} \ln^2\left(\frac{64}{r_{\min}^2 \epsilon^2}\right)$ in the same manner as in the proof of Theorem 2. The remaining part of the proof is almost the same as the proof of Theorem 2, and we omit the details.

## B.7 Bridging $\epsilon$, $\Delta_{\min}$ and $\Delta$.

In this subsection, we investigate the closed-form solution of $\Delta_{\min}$ in our formulation in some special cases of the fairness constraints. Based on the derived closed form expression, we provide evidences for the belief that the regret bound given by Theorem 3 is tighter than Theorem 2. Specifically, we show that when $m = 1$, $\Delta_{\min} = \epsilon(1 - \sum_i r_i)$, which immediately implies that the regret bound given by Theorem 3 is tighter than Theorem 2.

We also provide a better understanding of the relationship between the two optimality gaps $\Delta_{\min}$ and $\epsilon$ used in this paper and the traditional optimality gap $\Delta$ used in classical MAB setting. In fact, both $\Delta_{\min}$ and $\epsilon$ could be seen as a generalized version of $\Delta$, as both of them coincides with $\Delta$ under the classical MAB setting.

Consider the case when $m = 1$. In this special case, $\Delta_{\min}$ can be written in closed form, and we prove that $\Delta_{\min} \leq \epsilon$.

In fact, if $m = 1$, the extreme points of $\mathcal{D}$ are

$$
\mathcal{B} = \left\{ \begin{pmatrix} r_1 + 1 - \sum_i r_i \\ r_2 \\ \dots \\ r_N \end{pmatrix}, \begin{pmatrix} r_1 \\ r_2 + 1 - \sum_i r_i \\ \dots \\ r_N \end{pmatrix}, ...., \begin{pmatrix} r_1 \\ r_2 \\ \dots \\ r_N + 1 - \sum_i r_i \end{pmatrix} \right\}.
$$

Then for $\forall j \in [N]$, we have

$$
\left\langle \boldsymbol{\mu}, \begin{pmatrix} r_1 \\ r_2 \\ \dots \\ r_j + 1 - \sum_i r_i \\ \dots \\ r_N \end{pmatrix} \right\rangle = \langle \boldsymbol{\mu}, r \rangle + \mu_j \left(1 - \sum_i r_i\right)
$$

Hence $\max_{\boldsymbol{x} \in \mathcal{B}} \langle \boldsymbol{\mu}, x \rangle = \max_j \left( \langle \boldsymbol{\mu}, r \rangle + \mu_j (1 - \sum_i r_i) \right) = \langle \boldsymbol{\mu}, r \rangle + \mu^* (1 - \sum_i r_i)$, and $\Delta_{\min}$ can be written in closed form as

$$
\begin{aligned}
\Delta_{\min} &= \min_{\mu_j \neq \mu^*} \left( \langle \boldsymbol{\mu}, r \rangle + \mu^* (1 - \sum_i r_i) \right) - \left( \langle \boldsymbol{\mu}, r \rangle + \mu_j (1 - \sum_i r_i) \right) \\
&= \min_{\mu_j \neq \mu^*} (\mu^* - \mu_j)(1 - \sum_i r_i) = \Delta(1 - \sum_i r_i)
\end{aligned}
$$

On the other hand, when $m = 1$, $k = \min\{l \mid \sum_{i=1}^{l}(1 - r_{\sigma_i}) \geq 1 - \sum_{i=1}^{N} r_i, \ l \in \mathbb{Z}^+\} = 1$. Note that $\mu_{\sigma_1} = \mu^*$ due to the definition of $\sigma$, then

$$
\epsilon = \min_{\mu_i \neq \mu_{\sigma_1}} |\mu_i - \mu_{\sigma_1}| = \Delta.
$$

So we have $\Delta_{\min} = \Delta(1 - \sum_i r_i) < \Delta = \epsilon$, this implies that when $m = 1$, the regret bound in Theorem 3 is tighter than the regret bound in Theorem 2.

When $m > 1$, obtaining the closed form expression of $\Delta_{\min}$ is complicated, and it is hard to establish any direct inequality between the two regret bounds. However, based on the insights given by analyzing the case of $m = 1$, we believe that the regret bound in Theorem 3 is also likely to be tighter for general $m$ values.

We also note that, under the classical MAB setting, $m = 1$ and there is no fairness constraints, i.e., $\boldsymbol{r} = \boldsymbol{0}$, both $\Delta_{\min}$ and $\epsilon$ coincides with $\Delta$. This validates the reasonability and tightness of the two optimality gaps used in this paper.

## C  Proofs for Subsection 2.3

**Preliminary.** Since $\mathcal{D} = \{\boldsymbol{x} \mid \boldsymbol{x} \in [0,1]^N, ||\boldsymbol{x}||_1 \leq m, \ \boldsymbol{g}(\boldsymbol{x}) \leq \boldsymbol{0}\}$ is bounded, the function $g_k(\cdot), \ \forall k \in [K]$, has bounded function value on $\mathcal{D}$. In other words, there exists a constant $U > 0$ such that $g_k(\boldsymbol{x}) \leq U, \ \forall \boldsymbol{x} \in \mathcal{D}, \ k \in [K]$.

### C.1  Proof of Theorem 4

Firstly, we give a proof sketch of Theorem 4. The high-level idea of our proof for regret bound is decomposing the regret as follows,

$$Regret_T \leq T\boldsymbol{\mu} \cdot \boldsymbol{x}^* - E[\sum_{t=1}^{T} \boldsymbol{\mu} \cdot \boldsymbol{x}(t)]$$

$$= \underbrace{T\boldsymbol{\mu} \cdot (\boldsymbol{x}^* - \boldsymbol{a}^{\epsilon_t,*})}_{\text{term A}} + \underbrace{E[\sum_{t=1}^{T} (\hat{\boldsymbol{\mu}}(t) \cdot \boldsymbol{x}^{\epsilon_t,*} - \hat{\boldsymbol{\mu}}(t) \cdot \boldsymbol{a}(t))]}_{\text{term B}}$$

$$+ \underbrace{E[\sum_{t=1}^{T} (\boldsymbol{\mu} - \hat{\boldsymbol{\mu}}(t)) \cdot \boldsymbol{x}^{\epsilon_t,*}]}_{\text{term C}} + \underbrace{E[\sum_{t=1}^{T} (\hat{\boldsymbol{\mu}}(t) - \boldsymbol{\mu}) \cdot \boldsymbol{a}(t)]}_{\text{term D}},$$

where $\boldsymbol{x}^{\epsilon_t,*}$ is one of the optimal solutions to the "$\epsilon_t$-pessimistic" version of LP (2), i.e.,

$$\boldsymbol{x}^{\epsilon_t,*} = \max_{0 \leq \boldsymbol{x} \leq 1} \ \langle \boldsymbol{\mu}, \boldsymbol{x} \rangle, \text{ s.t. } \boldsymbol{g}(\boldsymbol{x}) + \epsilon_t \boldsymbol{I} \leq \boldsymbol{0}, \ ||\boldsymbol{x}||_1 \leq m.$$

To prove the regret bound in Theorem 4, we bound the term B using the Lyapunov-drift analysis [38, 29], etc. Term A could be bounded based on the intuition that adding $\epsilon_t$-tightness to the feasible region only incurs $O(\epsilon_t)$ loss in the optimal objective value. Terms C and D could be regarded as "reward mismatch" and we bound them using the similar analysis from [1, 42] based on the concentration bound.

For the constraint violations bound at any $\tau \leq T$, we firstly characterize it by the bound of $||\boldsymbol{Q}(\tau)||_1$. Then we use the drift property of $\boldsymbol{Q}(\cdot)$ and convergence time analysis from random process community to bound $e^{||\boldsymbol{Q}(\tau)||_1}$, which could be transformed into high-probability upper bound of $||\boldsymbol{Q}(\tau)||_1$ via Markov inequality.

Now we give a proof of Theorem 4. To obtain the regret bound, as mentioned before, we first decompose the regret expression as follows,

$$Regret_T \leq T\boldsymbol{\mu} \cdot \boldsymbol{x}^* - E[\sum_{t=1}^{T} \boldsymbol{\mu} \cdot \boldsymbol{a}(t)]$$

$$= \underbrace{T\boldsymbol{\mu} \cdot (\boldsymbol{x}^* - \boldsymbol{x}^{\epsilon_t,*})}_{\text{term A}} + \underbrace{E[\sum_{t=1}^{T} (\hat{\boldsymbol{\mu}}(t) \cdot \boldsymbol{x}^{\epsilon_t,*} - \hat{\boldsymbol{\mu}}(t) \cdot \boldsymbol{a}(t))]}_{\text{term B}} \qquad (26)$$

$$+ \underbrace{E[\sum_{t=1}^{T} (\boldsymbol{\mu} - \hat{\boldsymbol{\mu}}(t)) \cdot \boldsymbol{x}^{\epsilon_t,*}]}_{\text{term C}} + \underbrace{E[\sum_{t=1}^{T} (\hat{\boldsymbol{\mu}}(t) - \boldsymbol{\mu}) \cdot \boldsymbol{a}(t)]}_{\text{term D}}.$$

We next present a sequence of lemmas that bounds the terms above.

**Lemma 3** *The algorithm UCB-PLLP guarantees that*

$$E[\sum_{t=1}^{T} \langle \hat{\boldsymbol{\mu}}(t), \boldsymbol{x}^{\epsilon_t,*} - \boldsymbol{a}(t) \rangle] \leq \sum_{t=1}^{T} \alpha_t K(U^2 + \epsilon_t^2) + \sum_{t=1}^{T} \alpha_t(\epsilon_t - \delta) ||\boldsymbol{Q}(t)||_1 \mathbb{1}(\epsilon_t > \delta), \tag{27}$$

**Lemma 4** *For any sequence $\{\epsilon_t\}_t$ which satisfies $0 \leq \epsilon_t \leq \delta$, $\forall t$, we have that*

$$\sum_{t=1}^{T} \boldsymbol{\mu} \cdot (\boldsymbol{x}^* - \boldsymbol{x}^{\epsilon_t,*}) \leq m \sum_{t=1}^{T} \frac{\epsilon_t}{\delta}. \tag{28}$$

**Lemma 5** $\forall \boldsymbol{a} \in \mathcal{A}$, *the following inequality holds*

$$E[|\sum_{t=1}^{T} \boldsymbol{a} \cdot \hat{\boldsymbol{\mu}}(t) - \boldsymbol{a} \cdot \boldsymbol{\mu}|] \leq 4m\sqrt{T \ln T} + 4m. \tag{29}$$

Then according to Lemma 4, we bound term A as follows,

$$\text{term A} \leq m \sum_{t=1}^{T} \frac{\epsilon_t}{\delta}. \tag{30}$$

Term B could be upper-bounded based on the Lemma 3:

$$\text{term B} \leq \sum_{t=1}^{T} \alpha_t K(U^2 + \epsilon_t^2) + \sum_{t=1}^{T} \alpha_t(\epsilon_t - \delta) ||\boldsymbol{Q}(t)||_1 \mathbb{1}(\epsilon_t > \delta). \tag{31}$$

And Lemma 5 gives

$$\text{term C} \leq 4m\sqrt{T \ln T} + 4m, \ \text{term D} \leq 4m\sqrt{T \ln T} + 4m. \tag{32}$$

Thus, combine the (30), (31), and (32), we have the following upper-bound for regret,

$$Regret_T \leq m \sum_{t=1}^{T} \frac{\epsilon_t}{\delta} + 8m\sqrt{T \ln T} + 8m + \sum_{t=1}^{T} \alpha_t K(U^2 + \epsilon_t^2) + \sum_{t=1}^{T} \alpha_t(\epsilon_t - \delta) ||\boldsymbol{Q}(t)||_1 \mathbb{1}(\epsilon_t > \delta). \tag{33}$$

Let $p = \frac{\delta}{4}$ and $q = \frac{8N\sqrt{K}}{\delta}$. Since $\alpha_t = \frac{q}{\sqrt{t}}$ and $\epsilon_t = \frac{p}{\sqrt{t}}$, (33) yields

$$Regret_T \leq m \sum_{t=1}^{T} \frac{\epsilon_t}{\delta} + \sum_{t=1}^{T} \alpha_t K(U^2 + \epsilon_t^2) + \sum_{t=1}^{T} \alpha_t(\epsilon_t - \delta) ||\boldsymbol{Q}(t)||_1 \mathbb{1}(\epsilon_t > \delta) + 8m\sqrt{T \ln T} + 8m$$

$$\overset{(a)}{\leq} \frac{mp}{\delta} \sum_{t=1}^{T} \frac{1}{\sqrt{t}} + KU^2 q \sum_{t=1}^{T} \frac{1}{\sqrt{t}} + Kqp^2 \sum_{t=1}^{T} \frac{1}{t^{\frac{3}{2}}} + \sum_{t=1}^{\min\{\tau:\epsilon_\tau \leq \delta\}} \alpha_t \epsilon_t t(U + \epsilon_0) + 8m\sqrt{T \ln T} + 8m$$

$$\overset{(b)}{\leq} \frac{2mp}{\delta} \sqrt{T} + 2KU^2 q\sqrt{T} + 2Kqp^2 + 8m\sqrt{T \ln T} + 8m$$

$$\leq \frac{m}{2} \sqrt{T} + 16NK^{\frac{3}{2}} U^2 \frac{\sqrt{T}}{\delta} + NK\sqrt{K}\delta + 8m\sqrt{T \ln T} + 8m. \tag{34}$$

where (a) in (34) follows from the fact that $||\boldsymbol{Q}(t)||_1 \leq Ut + \sum_{\tau=1}^{t} \epsilon_\tau \leq t(U + \epsilon_0)$; (b) in (34) holds due to $\epsilon_t \leq \delta$, $\sum_{t=1}^{T} \frac{1}{\sqrt{t}} \leq 2\sqrt{T}$ and $\sum_{t=1}^{T} t^{-\frac{3}{2}} \leq 2$.

To obtain the constraint violations bound, according to the update rule of $\boldsymbol{Q}(t)$, we have

$$Q_k(t+1) \geq \sum_{t=1}^{T} g_k(\boldsymbol{a}(t)) + \sum_{t=1}^{T} \epsilon_t$$

$$\Rightarrow \text{Vio}_k(T) = \sum_{t=1}^{T} g_k(\boldsymbol{a}(t)) \leq Q_k(T+1) - \sum_{t=1}^{T} \epsilon_t, \ \forall k. \tag{35}$$

Let $\eta = \frac{\delta}{4[\max\{\delta/4, K(U+\epsilon_0)\}]^2 + \max\{\delta/4, K(U+\epsilon_0)\}\delta/3}$. Combine (35) with the Markov inequality gives

$$
\begin{aligned}
\Pr(\text{Vio}_k(T) \geq 0) &\leq \Pr(Q_k(T+1) \geq \sum_{t=1}^{T} \epsilon_t) \\
&\leq \Pr(||\boldsymbol{Q}(T+1)||_1 \geq \sum_{t=1}^{T} \epsilon_t) \leq \frac{E[e^{\frac{\eta}{\sqrt{K}}||\boldsymbol{Q}(T+1)||_1}]}{e^{\frac{\eta}{\sqrt{K}}\sum_{t=1}^{T} \epsilon_t}} \\
&\overset{(a)}{\leq} \frac{1 + \frac{8e^{\eta[\max\{\delta K(U+\epsilon_0), \frac{\delta^2}{4}\} + \frac{2}{\alpha_T}N + 2K(U^2+\epsilon_t^2)]}}{\eta\delta^2}}{e^{\frac{\eta}{\sqrt{K}}\sum_{t=1}^{T} \epsilon_T}} \\
&\leq e^{-\frac{\eta}{\sqrt{K}}\sum_{t=1}^{T} \epsilon_t} + \frac{8}{\eta\delta^2}e^{\eta[\max\{\delta K(U+\epsilon_0), \frac{\delta^2}{4}\} + \frac{2}{\lambda_T}N + 2K(U^2+\epsilon_t^2)] - \frac{\eta}{\sqrt{K}}\sum_{t=1}^{T} \epsilon_t} \\
&\leq e^{-\frac{\eta}{\sqrt{K}}\sum_{t=1}^{T} \epsilon_t} + \frac{8}{\eta\delta^2}e^{\eta[\max\{\delta K(U+\epsilon_0), \frac{\delta^2}{4}\} + \frac{2}{\alpha_T}N + 2K(U^2+\epsilon_0^2)] - \frac{\eta}{\sqrt{K}}\sum_{t=1}^{T} \epsilon_t} \\
&\leq e^{-\frac{\eta}{\sqrt{K}}\sum_{t=1}^{T} \epsilon_t} + \frac{8e^{\eta[\max\{\delta K(U+\epsilon_0), \frac{\delta^2}{4}\} + 2K(U^2+\epsilon_0^2)]}}{\eta\delta^2}e^{\frac{2}{\alpha_T}N\eta - \frac{\eta}{\sqrt{K}}\sum_{t=1}^{T} \epsilon_t} \\
&\leq [1 + \frac{8e^{\eta[\max\{\delta K(U+\epsilon_0), \frac{\delta^2}{4}\} + 2K(U^2+\epsilon_0^2)]}}{\eta\delta^2}]e^{\frac{2}{\alpha_T}N\eta - \frac{\eta}{\sqrt{K}}\sum_{t=1}^{T} \epsilon_t} = O(e^{-\delta\sqrt{T}})
\end{aligned}
\tag{36}
$$

where (a) in (36) comes from the following lemma:

**Lemma 6** *When $\epsilon \leq \frac{\delta}{4}$, UCB-PLLP guarantees that*

$$
E[e^{\frac{\eta}{\sqrt{K}}||\boldsymbol{Q}(t)||_1}] \leq E[e^{\eta||\boldsymbol{Q}(t)||}] \leq 1 + \frac{8e^{\eta[\max\{\delta K(U+\epsilon_0), \frac{\delta^2}{4}\} + \frac{2}{\alpha_t}N + 2K(U^2+\epsilon_t^2)]}}{\eta\delta^2},
\tag{37}
$$

*where $\eta = \frac{\delta}{4[\max\{\delta/4, K(U+\epsilon_0)\}]^2 + \max\{\delta/4, K(U+\epsilon_0)\}\delta/3}$.*

This completes the proof.

## C.2 Proof of Lemma 3

Proof Define Lyapunov drift $\Delta(t) = \frac{1}{2}(||\boldsymbol{Q}(t+1)||^2 - ||\boldsymbol{Q}(t)||^2)$. According to the evolution dynamics of $\boldsymbol{Q}(t)$, we have

$$
\begin{aligned}
\Delta(t) &= \frac{1}{2}[||\boldsymbol{Q}(t+1)||^2 - ||\boldsymbol{Q}(t)||^2] = [\boldsymbol{g}(\boldsymbol{a}(t)) + \epsilon_t \boldsymbol{I}]^T \boldsymbol{Q}(t) + \frac{1}{2}||\boldsymbol{g}(\boldsymbol{a}(t)) + \epsilon_t \boldsymbol{I}||^2 \\
&\overset{(a)}{\leq} [\boldsymbol{g}(\boldsymbol{a}(t)) + \epsilon_t \boldsymbol{I}]^T \boldsymbol{Q}(t) + K(U^2 + \epsilon_t^2),
\end{aligned}
\tag{38}
$$

where (a) in (38) holds due to $g_k(\cdot)$ is bounded by $U$. Recall that $\mathcal{A} = \{\boldsymbol{a}|\boldsymbol{a} \in \{0,1\}^N, ||\boldsymbol{a}||_1 \leq m\}$ and $\mathcal{D} = \{\boldsymbol{x}|\boldsymbol{x} \in [0,1]^N, ||\boldsymbol{x}||_1 \leq m, \boldsymbol{g}(\boldsymbol{x}) \leq \boldsymbol{0}\}$, and define $\hat{\mathcal{D}} = \{\boldsymbol{x}|\boldsymbol{x} \in [0,1]^N, ||\boldsymbol{x}||_1 \leq m\}$ then we have

$$
\begin{aligned}
\max_{\boldsymbol{a} \in \mathcal{A}} \left\langle \hat{\boldsymbol{\mu}}(t) - \alpha_t \sum_{k=1}^{K} \nabla g_k(\boldsymbol{a}(t-1)) Q_k(t), \boldsymbol{a} \right\rangle &= \max_{\boldsymbol{x} \in \hat{\mathcal{D}}} \left\langle \hat{\boldsymbol{\mu}}(t) - \alpha_t \sum_{k=1}^{K} \nabla g_k(\boldsymbol{a}(t-1)) Q_k(t), \boldsymbol{x} \right\rangle \\
&\geq \max_{\boldsymbol{x} \in \mathcal{D}} \left\langle \hat{\boldsymbol{\mu}}(t) - \alpha_t \sum_{k=1}^{K} \nabla g_k(\boldsymbol{a}(t-1)) Q_k(t), \boldsymbol{x} \right\rangle
\end{aligned}
$$

Since $\boldsymbol{a}(t) = \arg\max_{\boldsymbol{a} \in \mathcal{A}} \left\langle \hat{\boldsymbol{\mu}}(t) - \alpha_t \sum_{k=1}^K \nabla g_k(\boldsymbol{a}(t-1)) Q_k(t), \boldsymbol{a} \right\rangle$, then for any $\boldsymbol{y} \in \mathcal{D}$ we have

$$\left\langle \hat{\boldsymbol{\mu}}(t) - \alpha_t \sum_{k=1}^K \nabla g_k(\boldsymbol{a}(t-1)) Q_k(t), \boldsymbol{a}(t) \right\rangle \geq \left\langle \hat{\boldsymbol{\mu}}(t) - \alpha_t \sum_{k=1}^K \nabla g_k(\boldsymbol{a}(t-1)) Q_k(t), \boldsymbol{y} \right\rangle$$

$$\Leftrightarrow \langle \hat{\boldsymbol{\mu}}(t), \boldsymbol{a}(t) \rangle - \alpha_t \sum_{k=1}^K Q_k(t) \langle \nabla g_k(\boldsymbol{a}(t-1)), a(t) \rangle \geq \langle \hat{\boldsymbol{\mu}}(t), \boldsymbol{y} \rangle - \alpha_t \sum_{k=1}^K Q_k(t) \langle \nabla g_k(\boldsymbol{a}(t-1)), \boldsymbol{y} \rangle$$

$$\Leftrightarrow \langle \hat{\boldsymbol{\mu}}(t), \boldsymbol{a}(t) \rangle - \alpha_t \sum_{k=1}^K Q_k(t) \langle \nabla g_k(\boldsymbol{a}(t-1)), \boldsymbol{a}(t) - \boldsymbol{a}(t-1) \rangle \geq \langle \hat{\boldsymbol{\mu}}(t), \boldsymbol{y} \rangle - \alpha_t \sum_{k=1}^K Q_k(t) \langle \nabla g_k(\boldsymbol{a}(t-1)), \boldsymbol{y} - \boldsymbol{a}(t-1) \rangle$$

$$\overset{(a)}{\Leftrightarrow} \langle \hat{\boldsymbol{\mu}}(t), \boldsymbol{a}(t) \rangle - \alpha_t \sum_{k=1}^K Q_k(t) g_k(\boldsymbol{a}(t)) \geq \langle \hat{\boldsymbol{\mu}}(t), \boldsymbol{y} \rangle - \alpha_t \sum_{k=1}^K Q_k(t) g_k(\boldsymbol{y})$$

$$\Leftrightarrow \langle \hat{\boldsymbol{\mu}}(t), \boldsymbol{a}(t) \rangle - \alpha_t [\boldsymbol{g}(\boldsymbol{a}(t))]^T \boldsymbol{Q}(t) \geq \langle \hat{\boldsymbol{\mu}}(t), \boldsymbol{y} \rangle - \alpha_t [\boldsymbol{g}(\boldsymbol{y})]^T \boldsymbol{Q}(t)$$

where (a) holds since $g_k(\cdot)$ is linear. The above inequality is also equivalent to:

$$\langle \hat{\boldsymbol{\mu}}(t), \boldsymbol{y} \rangle - \alpha_t [\boldsymbol{g}(\boldsymbol{y}) + \epsilon_t \boldsymbol{I}]^T \boldsymbol{Q}(t) \leq \langle \hat{\boldsymbol{\mu}}(t), \boldsymbol{x}(t) \rangle - \alpha_t [\boldsymbol{g}(\boldsymbol{a}(t)) + \epsilon_t \boldsymbol{I}]^T \boldsymbol{Q}(t) \tag{39}$$

Divide both sides of (39) by $\alpha_t$ and add it to inequality (38), then we obtain

$$\Delta(t) + \frac{1}{\alpha_t} \langle \hat{\boldsymbol{\mu}}(t), \boldsymbol{y} \rangle - [\boldsymbol{g}(\boldsymbol{y}) + \epsilon_t \boldsymbol{I}]^T \boldsymbol{Q}(t) \leq \frac{1}{\alpha_t} \langle \hat{\boldsymbol{\mu}}(t), \boldsymbol{a}(t) \rangle + K(U^2 + \epsilon_t^2). \tag{40}$$

Rearrange terms yields

$$\Delta(t) \leq \frac{1}{\alpha_t} \langle \hat{\boldsymbol{\mu}}(t), \boldsymbol{a}(t) - \boldsymbol{y} \rangle + K(U^2 + \epsilon_t^2) + [\boldsymbol{g}(\boldsymbol{y}) + \epsilon_t \boldsymbol{I}]^T \boldsymbol{Q}(t). \tag{41}$$

Denote $\boldsymbol{H}(t) = [\boldsymbol{Q}(t), \hat{\boldsymbol{\mu}}(t)]$ the current state. Substitute $\boldsymbol{y}$ by $\boldsymbol{x}^{\epsilon_t, *}$ in (41) and conditional on $\boldsymbol{H}(t) = \boldsymbol{h}$, we have

$$E[\Delta(t) | \boldsymbol{H}(t) = \boldsymbol{h}] \leq \frac{1}{\alpha_t} E[\langle \hat{\boldsymbol{\mu}}(t), \boldsymbol{a}(t) - \boldsymbol{x}^{\epsilon_t, *} \rangle | \boldsymbol{H}(t) = \boldsymbol{h}] + K(U^2 + \epsilon_t^2) + [\boldsymbol{g}(\boldsymbol{x}^{\epsilon_t, *}) + \epsilon_t \boldsymbol{I}]^T \boldsymbol{Q}(t)$$

$$\overset{(a)}{\leq} [\boldsymbol{g}(\boldsymbol{x}^{\epsilon_t, *}) + \epsilon_t \boldsymbol{I}]^T \boldsymbol{Q}(t) \mathbb{I}(\epsilon_t \leq \delta) + [\boldsymbol{g}(\boldsymbol{x}^{\epsilon_t, *}) + \delta \boldsymbol{I}]^T \boldsymbol{Q}(t) \mathbb{I}(\epsilon_t > \delta)$$

$$+ \frac{1}{\alpha_t} E[\langle \hat{\boldsymbol{\mu}}(t), \boldsymbol{a}(t) - \boldsymbol{x}^{\epsilon_t, *} \rangle | \boldsymbol{H}(t) = \boldsymbol{h}] + K(U^2 + \epsilon_t^2) + (\epsilon_t - \delta) \|\boldsymbol{Q}(t)\|_1 \mathbb{I}(\epsilon_t > \delta)$$

$$\overset{(b)}{\leq} \frac{1}{\alpha_t} E[\langle \hat{\boldsymbol{\mu}}(t), \boldsymbol{a}(t) - \boldsymbol{x}^{\epsilon_t, *} \rangle | \boldsymbol{H}(t) = \boldsymbol{h}] + K(U^2 + \epsilon_t^2) + (\epsilon_t - \delta) \|\boldsymbol{Q}(t)\|_1 \mathbb{I}(\epsilon_t > \delta),$$

$$\tag{42}$$

where (a) and (b) in (42) come from the fact that $\boldsymbol{g}(\boldsymbol{x}^{\epsilon_t, *}) + \epsilon_t \boldsymbol{I} \leq 0$. Thus, the Lyapunov drift $\Delta(t)$ satisfies

$$E[\Delta(t) | \boldsymbol{H}(t) = \boldsymbol{h}] \leq \frac{1}{\alpha_t} E[\langle \hat{\boldsymbol{\mu}}(t), \boldsymbol{a}(t) - \boldsymbol{x}^{\epsilon_t, *} \rangle | \boldsymbol{H}(t) = \boldsymbol{h}] + K(U^2 + \epsilon_t^2) + (\epsilon_t - \delta) \|\boldsymbol{Q}(t)\|_1 \mathbb{I}(\epsilon_t > \delta). \tag{43}$$

Take expectations with respect to $\boldsymbol{H}(t)$, multiply $\alpha_t$ on both sides and conduct a telescope summing over $t$ lead to

$$E[\sum_{t=1}^T \langle \hat{\boldsymbol{\mu}}(t), \boldsymbol{x}^{\epsilon_t, *} - \boldsymbol{a}(t) \rangle]$$

$$\overset{(a)}{\leq} \alpha_1 E[\|\boldsymbol{Q}(1)\|^2] - \alpha_T E[\|\boldsymbol{Q}(T+1)\|^2] + \sum_{t=1}^T \alpha_t K(U^2 + \epsilon_t^2) + \sum_{t=1}^T \alpha_t (\epsilon_t - \delta) \|\boldsymbol{Q}(t)\|_1 \mathbb{I}(\epsilon_t > \delta)$$

$$\overset{(b)}{\leq} \sum_{t=1}^T \alpha_t K(U^2 + \epsilon_t^2) + \sum_{t=1}^T \alpha_t (\epsilon_t - \delta) \|\boldsymbol{Q}(t)\|_1 \mathbb{I}(\epsilon_t > \delta),$$

$$\tag{44}$$

where (a) in (44) holds since $\alpha_t$ is non-increasing over time $t$; (b) in (44) holds due to $\|\boldsymbol{Q}(1)\| = 0$. Then we complete the proof.

## C.3 Proof of Lemma 4

Proof Due to the Slater condition, there exists a $\delta > 0$ and $\hat{x} \in \mathcal{D}$ such that $g(\hat{x}) \leq -\delta I$. Define $x^{\epsilon_t} = (1 - \frac{\epsilon_t}{\delta})x^* + \frac{\epsilon_t}{\delta}\hat{x}$. Since $0 \leq \epsilon_t \leq \delta$ and

$$||x^{\epsilon_t}||_1 = (1 - \frac{\epsilon_t}{\delta})||x^*||_1 + \frac{\epsilon_t}{\delta}||\hat{x}||_1 = m(1 - \frac{\epsilon_t}{\delta}) + \frac{\epsilon_t}{\delta}m = m,$$

we can verify that $x^{\epsilon_t} \in \mathcal{D}$ and

$$g(x^{\epsilon_t}) = g((1 - \frac{\epsilon_t}{\delta})x^* + \frac{\epsilon_t}{\delta}\hat{x}) = (1 - \frac{\epsilon_t}{\delta})g(x^*) + \frac{\epsilon_t}{\delta}g(\hat{x}) \leq 0 - \epsilon_t I = -\epsilon_t I, \tag{45}$$

where the second equality in (45) holds due to the linearity of $g(\cdot)$. Hence $x^{\epsilon_t}$ is exactly the feasible solution to the "$\epsilon_t$-tightness" version of the optimization problem (2) and hence we have $\mu \cdot x^{\epsilon_t} \leq \mu \cdot x^{\epsilon_t,*}$. So we can obtain

$$\sum_{t=1}^{T} \mu \cdot (x^* - x^{\epsilon_t,*}) \leq \sum_{t=1}^{T} \mu \cdot (x^* - x^{\epsilon_t})$$

$$\leq \sum_{t=1}^{T} \mu \cdot (x^* - (1 - \frac{\epsilon_t}{\delta})x^* - \frac{\epsilon_t}{\delta}\hat{x}) \leq \sum_{t=1}^{T} \mu \cdot (x^* - (1 - \frac{\epsilon_t}{\delta})x^*) \tag{46}$$

$$= \sum_{t=1}^{T} \mu \cdot (\frac{\epsilon_t}{\delta}x^*) \leq m \sum_{t=1}^{T} \frac{\epsilon_t}{\delta}.$$

This completes the proof.

## C.4 Proof of Lemma 5

Proof We prove the Lemma 5 by following the standard analysis from the traditional bandits community. First we have the following supportive lemma.

**Lemma 7** *(Hoeffding's lemma) With probability at least $1 - 2T^{-1}$, we have*

$$|\hat{\mu}_i(t) - \mu_i| \leq \sqrt{\frac{2\ln T}{N_i(t)}}. \tag{47}$$

By the definition of $\hat{\mu}(t)$, $\forall a \in \mathcal{A}$ we have

$$|\sum_{t=1}^{T} a \cdot \hat{\mu}(t) - a \cdot \mu| \leq |\sum_{i=1}^{N}\sum_{t=1}^{T} a_i(\overline{\mu}_i(t) - \mu_i)| + \sum_{i=1}^{N}\sum_{t=1}^{T}\sqrt{\frac{2\ln t}{N_i(t)}}a_i \tag{48}$$

Using Lemma 7, we have that with probability $1 - 2T^{-1}$,

$$|\sum_{i=1}^{N}\sum_{t=1}^{T} a_i(\overline{\mu}_i(t) - \mu_i)| \leq \sum_{i=1}^{N}|\sum_{t=1}^{T} a_i(\overline{\mu}_i(t) - \mu_i)| \leq \sum_{a_i>0}\sum_{t=1}^{T}|\overline{\mu}_i(t) - \mu_i| \leq \sum_{a_i>0}\sum_{t=1}^{T}\sqrt{\frac{2\ln T}{N_i(t)}} \tag{49}$$

Combine (48) with (49), then with $1 - 2T^{-1}$ we obtain

$$|\sum_{t=1}^{T} a \cdot \hat{\mu}(t) - a \cdot \mu| \leq \sum_{a_i>0}\sum_{t=1}^{T}\sqrt{\frac{2\ln T}{N_i(t)}} + \sum_{i=1}^{N}\sum_{t=1}^{T}\sqrt{\frac{2\ln t}{N_i(t)}}a_i$$

$$\leq 2\sum_{a_i>0}\sum_{t=1}^{T}\sqrt{\frac{2\ln T}{N_i(t)}} \overset{(a)}{\leq} 2\sqrt{2\ln T}\sum_{a_i>0}\sqrt{2N_i(T)} \leq 4m\sqrt{T\ln T} \tag{50}$$

where (a) in (50) comes from the fact that $\sum_{i=1}^{t}\frac{1}{\sqrt{i}} \leq 2\sqrt{t}$. Therefore, we can get

$$E[|\sum_{t=1}^{T} a \cdot \hat{\mu}(t) - a \cdot \mu|] \leq (1 - \frac{2}{T})4m\sqrt{T\ln T} + \frac{2}{T}T \cdot 2m \leq 4m\sqrt{T\ln T} + 4m. \tag{51}$$

where the first inequality in (51) holds since $|a \cdot \hat{\mu}(t) - a \cdot \mu| \leq ||x||_1||\hat{\mu}(t) - \mu||_\infty \leq 2m$. And we complete the proof.

## C.5 Proof of Lemma 6

Proof According to (41), for any $\boldsymbol{y} \in \mathcal{D}$, we have

$$E[\Delta(t)|\boldsymbol{H}(t) = \boldsymbol{h}]$$

$$= [\boldsymbol{g}(\boldsymbol{y}) + \epsilon_t \boldsymbol{I}]^T \boldsymbol{Q}(t) - \frac{1}{\alpha_t} E[\langle \hat{\boldsymbol{\mu}}(t), \boldsymbol{a}(t) - \boldsymbol{y} \rangle | \boldsymbol{H}(t) = \boldsymbol{h}] + K(U^2 + \epsilon_t^2) \tag{52}$$

$$\leq [\boldsymbol{g}(\boldsymbol{y}) + \epsilon_t \boldsymbol{I}]^T \boldsymbol{Q}(t) + \frac{1}{\alpha_t} E[\langle \hat{\boldsymbol{\mu}}(t), \boldsymbol{y} \rangle | \boldsymbol{H}(t) = \boldsymbol{h}] + K(U^2 + \epsilon_t^2)$$

Substitute $\boldsymbol{y}$ by $\hat{\boldsymbol{x}}$ into (52) we get

$$E[\Delta(t)|\boldsymbol{H}(t) = \boldsymbol{h}]$$

$$\leq [\boldsymbol{g}(\hat{\boldsymbol{x}}) + \epsilon_t \boldsymbol{I}]^T \boldsymbol{Q}(t) + \frac{1}{\alpha_t} E[\langle \hat{\boldsymbol{\mu}}(t), \hat{\boldsymbol{x}} \rangle | \boldsymbol{H}(t) = \boldsymbol{h}] + K(U^2 + \epsilon_t^2)$$

$$\overset{(a)}{\leq} [-\delta \boldsymbol{I} + \epsilon_t \boldsymbol{I}]^T \boldsymbol{Q}(t) + \frac{1}{\alpha_t} N + K(U^2 + \epsilon_t^2) \tag{53}$$

$$= -(\delta - \epsilon_t)||\boldsymbol{Q}(t)||_1 + \frac{1}{\alpha_t} N + K(U^2 + \epsilon_t^2)$$

where (a) in (53) holds due to the Slater condition. (53) implies that

$$E[||\boldsymbol{Q}(t+1)||^2 - ||\boldsymbol{Q}(t)||^2 | \boldsymbol{H}(t) = \boldsymbol{h}] \leq -2(\delta - \epsilon_t)||\boldsymbol{Q}(t)||_1 + \frac{2}{\alpha_t} N + 2K(U^2 + \epsilon_t^2) \tag{54}$$

Next we define $\tilde{L}(t) = ||\boldsymbol{Q}(t)||$, and when $\tilde{L}(t) \geq \frac{\frac{2}{\alpha_t} N + 2K(U^2 + \epsilon_t^2)}{\delta}$, we have

$$E[||\boldsymbol{Q}(t+1)|| - ||\boldsymbol{Q}(t)|| | \boldsymbol{H}(t) = \boldsymbol{h}]$$

$$= E[\sqrt{||\boldsymbol{Q}(t+1)||^2} - \sqrt{||\boldsymbol{Q}(t)||^2} | \boldsymbol{H}(t) = \boldsymbol{h}]$$

$$\leq \frac{1}{2||\boldsymbol{Q}(t)||} E[||\boldsymbol{Q}(t+1)||^2 - ||\boldsymbol{Q}(t)||^2 | \boldsymbol{H}(t) = \boldsymbol{h}]$$

$$\overset{(a)}{\leq} -(\delta - \epsilon_t) \frac{||\boldsymbol{Q}(t)||_1}{||\boldsymbol{Q}(t)||} + \frac{\frac{1}{\alpha_t} N + K(U^2 + \epsilon_t^2)}{||\boldsymbol{Q}(t)||} \tag{55}$$

$$\leq -(\delta - \epsilon_t) + \frac{\frac{1}{\alpha_t} N + K(U^2 + \epsilon_t^2)}{||\boldsymbol{Q}(t)||}$$

$$\leq -(\delta - \epsilon_t) + \frac{\delta}{2} = -(\frac{\delta}{2} - \epsilon_t) \overset{(b)}{\leq} -\frac{\delta}{4},$$

where (a) in (55) follows from (54); (b) in (55) is due to $\epsilon_t \leq \frac{\delta}{4}$. Besides, based on the update rule of $\boldsymbol{Q}(t)$, the following inequality holds,

$$||\boldsymbol{Q}(t+1)|| - ||\boldsymbol{Q}(t)|| \leq ||\boldsymbol{Q}(t+1) - \boldsymbol{Q}(t)|| \leq K(U + \epsilon_t) \leq K(U + \epsilon_0), \forall t. \tag{56}$$

Define $\eta = \frac{\delta}{4[\max\{\delta/4, K(U+\epsilon_0)\}]^2 + \max\{\delta/4, K(U+\epsilon_0)\}\delta/3}$, then we can derive that

$$E[e^{\eta||\boldsymbol{Q}(t)||}] \leq 1 + \frac{8e^{\eta[\max\{\delta K(U+\epsilon_0), \frac{\delta^2}{4}\} + \frac{2}{\alpha_t} N + 2K(U^2 + \epsilon_t^2)]}}{\eta \delta^2}, \tag{57}$$

according to the lemma below.

**Lemma 8** *(Lemma 3.8 in [2]) Let $\boldsymbol{S}(t)$ be the state of Markov chain, $L(t)$ be a Lyapunov function and its drift denotes $\Delta(t) = L(t+1) - L(t)$. When the following two conditions satisfied*

- *There exists constant $\gamma > 0$, increasing sequence $\{\theta\}_{t=1}^T$ such that $E[\Delta(t)|\boldsymbol{S}(t) = \boldsymbol{s}] \leq -\gamma$ when $L(t) \geq \theta_t$.*

- *$||L(t+1) - L(t)|| \leq v$ holds for any $t$.*

*Then we have*

$$E[e^{\eta L(t)}] \leq e^{\eta L(0)} + \frac{2e^{\eta(\max\{v, \gamma\} + \theta_t)}}{\eta \gamma}, \tag{58}$$

*where $\eta = \frac{\gamma}{[\max\{v, \gamma\}]^2 + \max\{v, \gamma\}\gamma/3}$.*

Since $||\boldsymbol{Q}(t)||_1 \le \sqrt{K}||\boldsymbol{Q}(t)||$, according to (57) it is obvious that

$$E[e^{\frac{\eta}{\sqrt{K}}||\boldsymbol{Q}(t)||_1}] \le 1 + \frac{8e^{\eta[\max\{\delta K(U+\epsilon_0),\frac{\delta^2}{4}\}+\frac{2}{\alpha_t}N+2K(U^2+\epsilon_t^2)]}}{\eta\delta^2}, \tag{59}$$

Then we complete the proof.

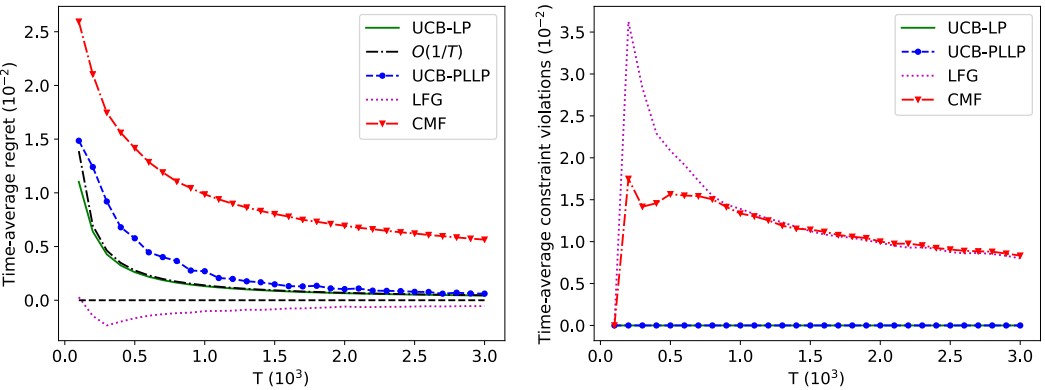

Figure 1: Results for fairness constraints

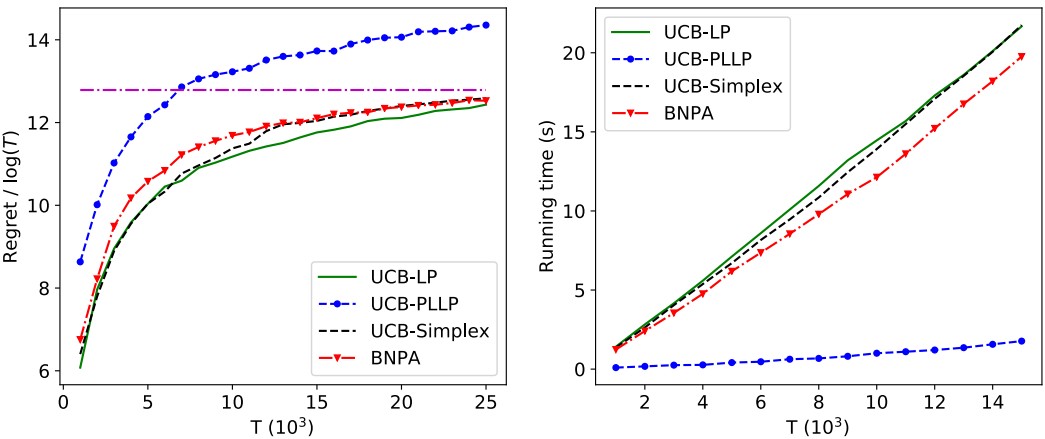

Figure 2: Results for knapsacks constraints

## D    Numerical experiments

In this section, we conduct numerical experiments for fairness constraints and knapsacks constraints, respectively, to validate the theoretical guarantees of our algorithms.

### D.1    Fairness constraints

For the simulation setup, we set $N = 4$, $m = 2$, $\boldsymbol{\mu} = (0.4, 0.5, 0.6, 0.7)$ and $\boldsymbol{r} = (0.5, 0.6, 0.4, 0.3)$. The representative baselines we used are: LFG [29] with $\eta = O(\frac{1}{\sqrt{T}})$, and CMF [49] with $\alpha = \infty$. We do not compare Thompson-Sampling-based algorithm [20] here as it requires additional knowledge on the prior distributions of $\boldsymbol{\mu}$ and the latent distributions of the arms, which is not assumed in other algorithms. In fact, their algorithm is the same as LFG except for the way of estimating rewards, which leads to almost the same performance. We do not compare [40, 12, 16] as their algorithms only work for $m = 1$. For fair comparisons, we use the same confidence bound in all algorithms. We simulate our algorithms and baselines for $T = 3 \times 10^4$ rounds. Every point in the figure is averaged over 100 independent trials.

**Results and analysis.** Figure 1 shows the time-averaged regret and total constraint violations of all compared algorithms. From this figure, we can see that UCB-LP indeed guarantees a constant

regret bound and zero constraint violations in expectation, which validates our theoretical results. The empirical performance of UCB-PLLP in our experiment also coincides with the results of Theorem 3.1. However, UCB-PLLP achieves a regret that is closed to constant in our experiment. It is not strange that LFG achieves negative regret as it compromises constraint violation to pull high-reward arms. For CMF, the empirical performance also matches its theoretical results.

## D.2 Knapsacks constraints

For the simulation setup, we choose $N = 5$ and $K = 3$. The mean rewards and mean costs of other arms are generated as follows: (a) $\mu_1 = 0.5$, $\lambda_{1,j} = 0.45$, $\forall j \in [3]$; (b) for all arms $2 \leq x \leq 5$, $\mu_x$ is sampled from Uniform($[0.5 - 2\sigma, 0.5 - \sigma]$), and $\lambda_{x,j}$ is sampled from Uniform($[0.45 + \sigma, 0.45 + 2\sigma]$), $\forall j \in [3]$, where $\sigma = 0.2$. We also set $B_j = 0.45T$, $\forall j \in [d]$. The baselines we used are these works which obtained logarithmic regret for deterministic costs: UCB-Simplex [18], and BNPA [41] with $\epsilon = O(\frac{\log T}{T})$. We do not compare UCB-Simplex-v2 [18] here as it is the same as BNPA with $\epsilon = 0$ (but when $\epsilon > 0$, the regret upper bound of BNPA is better than it). We do not compare [43, 15] as their algorithms and results require restrictive assumptions like only one resource. Since all baselines only work for $m = 1$, we set $m = 1$ in our simulation setup. All algorithms are simulated on the same datasets. Every point in every figure is averaged over 50 independent trials.

**Results and analysis.** Since all algorithms can guarantee zero constraint violations in expectation, we do not show the comparison of their constraint violations. Figure 2 (a) shows that the ratio of regret to $\log T$ of all algorithms except UCB-PLLP approaches a constant, i.e., achieving a logarithmic regret. This is consistent with their theoretical results. We can also see from this figure that UCB-LP has a best empirical performance under our experimental setup. Figure 2 (b) presents the running time of these algorithms, which highlights the advantage of UCB-PLLP in computational complexity.