# OpenReview forum: "Combinatorial Bandits with Linear Constraints: Beyond Knapsacks and Fairness"
_NeurIPS.cc/2022/Conference — NeurIPS 2022 Accept_

### Official Review · Reviewer_Rm7C · 2022-07-08

**Rating:** 7
**Confidence:** 3
**Soundness:** 2 fair
**Presentation:** 2 fair
**Contribution:** 1 poor

**Summary:**


The paper studies a combinatorial multi-armed bandits with linear long-term constraints, where the combinatorial constraints are cardinality constraints and the linear constraints are deterministic and known. The paper proposes an upper-confidence bound LP-style algorithm for this problem that provides a logarithmic instance-dependent regret and no constraint violations in expectation. Then, the authors develop a second algorithm with better running time, a $\tilde O(\sqrt{T})$ regret, and no constraints violation in high probability.

**Questions:**

Let me know if I’m missing something about the relation between classical bandit problems and your framework.
For instance, are there some previous works in which neither condition 1) nor condition 2) is satisfied?




**Limitations:**

The authors have adequately addressed the limitations and potential negative societal impact of their work

**Strengths And Weaknesses:**

I have a huge concern about the meaningfulness of the problem studied in the paper.
I think that the study of BwK is meaningful when at least one of the following condition is satisfied:
1) the costs are unknown,
2) the costs are not in expectation, i.e., if the regret minimizer chooses a vector x the actual suffered cost is not g(x) but g(a) where a is sampled from x. Then, if the budget ends the game is over.

If none of the conditions is satisfied, the problem reduces to a classical Linear bandit.
In particular, any no-regret algorithm for linear bandits with action space defined by the constraints in Problem (4) does satisfies the constraints in expectation and provides no-regret.
Notice that your algorithm in Theorem 1 follows this approach: it satisfies the constraints at each round (in expectation).


Minor comments:

The abstract does not specify all the assumptions in the framework. For instance, I suggest to specify in the abstract that the constraints are deterministic and time invariant, while the combinatorial constraints are cardinality constraints.

At line 42. I don’t see a clear relation with the result in [4]. In [4] the constraints are not deterministic and it could happen that the algorithm have to stop before round T since it ended the budget. For this reason $OPT(T)$ could be strictly smaller than $T\ OPT_{LP}$.
I’m quite sure that $OPT(T)=T \ OPT_{LP}$ with deterministic constraints that need to be satisfied in expectation, since the algorithm never stop.


Post rebuttal update: I'm completely convinced by the authors' answers.

---

> ### Author Response · Authors · 2022-08-02
> **To Reviewer Rm7C**
>
> Thanks for your comments. Our clarification to your concerns is below.
>
>
> ### Our problem can be reduced to linear bandits.
> Admittedly, such a reduction is plausible, but it fails in two senses.
> * Such a reduction requires solving an NP-hard problem at every round. In fact, similar discussions about this topic have already been carried out in [Rejwan and Mansour, 2020], which noted that combinatorial bandits can be reduced to linear bandits and solved by the well-established linear bandit algorithm LinUCB [Dani et al., 2008]. However, they pointed out that applying LinUCB to the combinatorial setting involves solving an NP-hard optimization problem. In contrast, UCB-LP has a much lower computational complexity than the reduced LinUCB algorithm since it is polynomial-time computable.
>
>
> * The reduction to linear bandits is also sub-optimal in the regret bound.
> Admittedly, any no-regret algorithm for linear bandits applies to our setting. However, such an algorithm would fail to achieve a logarithmic regret bound of $O(mN\log T)$ as our algorithm UCB-LP does [Lattimore and Szepesv´ari, 2020]. Furthermore, UCB-LP can guarantee a bounded regret for fairness setting. These facts justify the meaningfulness of our work. As noted by reviewer 7wPF and SVeo, the key novelty of our work lies in the regret analysis. Intuitively, the reduction from linear bandits to our setting ignores the structural properties of the constraints (e.g., concentration property bonus caused by the fairness constraints) and observations about each individual pulled arm, hence produces sub-optimal results.
>
>
>
>
> ### The constraints are deterministic and only need to be satisfied in expectation.
>  We remark that there is a large body of literature (e.g., [Li et al., 2019], [Xu et al., 2020]) on bandits with fairness constraints wherein the constraint functions are deterministic and the long-term fairness constraints are only required to be satisfied in expectations.  We extend the fairness constraints along this line to the general linear constraints that include other long-term constraint types, such as budget (knapsack) constraints. **In fact, even for budgeted-constrained online learning problems, many previous studies (e.g., [Jenatton et al., 2016], [Liakopoulos et al., 2019]) consider the deterministic costs and only require the long-term budget constraints to be satisfied asymptotically**, i.e., sublinear constraint violations. In our model, UCB-LP trivially satisfies this objective with high-probability since
>
> \begin{equation}
> \begin{split}
>     \sum_{t=1}^T {g}({a}(t)) = \sum_{t=1}^T ( {g}({a}(t)) - {g}({x}(t)) ) + \sum_{t=1}^T {g}({x}(t)) \le \sum_{t=1}^T ( {g}({a}(t)) - {g}({x}(t)) )
> \end{split}
> \end{equation}
> \begin{equation}
> \begin{split}
> = \sum_{t=1}^T {g}({a}(t)-{x}(t)) = {g} ( \sum_{t=1}^T ({a}(t)-{x}(t))  ) \overset{\text{Azuma inequality}}{\le} (\text{high-prob}) O(\sqrt{T}) \cdot {1}_N
> \end{split}
> \end{equation}
> ### Our algorithm UCB-PLLP can guarantee the constraints stringently.
> Although the constraints satisfaction for UCB-LP only holds in expectation, it is worth mentioning that our another Lyapunov-based algorithm UCB-PLLP can satisfy the constraints stringently with high-probability and achieve the (near) optimal problem-independent regret guarantee.
>
>
> ### OPT= T$\cdot$OPT$_{\text{LP}}$ holds in our case.
> Thanks for the insight you provided. Maybe the intuition of $\text{OPT}= T\cdot \text{OPT}_{\text{LP}}$ is clear when the deterministic constraints that need to be satisfied in expectation, proving it rigorously is non-trivial in my opinion.
>
> **We sincerely hope the reviewer will reconsider the rating based on our response.**
>
> ### References:
> [Rejwan and Mansour, 2020]: Top-k combinatorial bandits with fullbandit feedback
>
> [Dani et al., 2008]: Stochastic linear optimization under
> bandit feedback
>
> [Lattimore and Szepesv´ari, 2020]: Bandit algorithms
>
> [Jenatton et al., 2016]: Adaptive algorithms for online convex optimization with long-term constraints.
>
> [Liakopoulos et al., 2019]: Online optimization with long-term budget constraints
>
> [Li et al., 2019]: Combinatorial sleeping bandits with fairness constraints
>
> [Xu et al., 2020]: Combinatorial multi-armed bandits with concave rewards and fairness constraints

---

> ### Author Response · Authors · 2022-08-08
> **To Reviewer Rm7C**
>
> Here we explain in more detail why the highly established algorithm for linear bandits (e.g., LinUCB) is time-expensive in our model. Specifically, to apply into our model, at each round LinUCB needs to involve an optimization problem to yield an extreme point and the optimistic estimate of $\mu$:
> \begin{equation}
> (x(t),\hat{\mu}(t))=\arg\max_{   (x,\theta)\in \mathcal{D}\times C_t     } \langle x,\theta \rangle,    (1)
> \end{equation}
> where $C_t$ is the confidence set of ${\mu}$ at round $t$. However, this optimization is NP-hard even for the unconstrained setting, i.e., $g(\cdot)=0$. Specifically, when $g(\cdot)=0$, (1) is a typical combinatorial optimization problem (since  $(x(t),\hat{\mu}(t))=\arg\max_{   (x,\theta)\in \mathcal{D}\times C_t     } \langle x,\theta \rangle=\arg\max_{   (x,\theta)\in \mathcal{A}\times C_t     } \langle a,\theta \rangle$ for $g(\cdot)=0$) and cannot be solved in a polynomial time. Thus, the running time of LinUCB is less likely to be accepted when $g(\cdot)$ is more complex. In fact, this computational concern has been pointed out in [Rejwan and Mansour, 2020] that LinUCB is unappealing for combinatorial bandits although they are actually a special case of linear bandits. While our algorithm UCB-LP only involves a polynomial-time-computable LP solving since we can maintain a UCB estimate for each coordinate of $\mu$ due to the semi-bandit feedback model.

---

> ### Author Response · Authors · 2022-08-10
> **To Reviewer Rm7C**
>
> Although hope we have addressed all of your main concerns, any further comments or questions are welcomed! Thank you for your time!

---

### Official Review · Reviewer_SVeo · 2022-07-09

**Rating:** 8
**Confidence:** 4
**Soundness:** 3 good
**Presentation:** 3 good
**Contribution:** 3 good

**Summary:**

The paper studies a general formulation of combinatorial bandits with linear constraints, which contains bandits with fairness and knapsacks as special cases. The authors propose and analyze an algorithm that combines UCB-based estimation with a linear program solution to yield logarithmic regret and zero constraint violation performance. Authors also propose a low-complexity variant of the algorithm that utilizes virtual queues to iteratively solve the LP component of the original design.

**Questions:**

The examples in the discussion are limited. Please expand on them to clarify how the formulation fits into the generic form.



**Limitations:**

I have not seen a discussion on limitations of this work. The emphasis has been on its contributions, which is understandable. It would be good to note any limitations that the authors recognize.

**Strengths And Weaknesses:**

The work provides a solid contribution to the literature on constrained MABs by encapsulating knapsack and fairness constraints within a unified formulation. The design and analysis are valuable contributions to this literature.

The novelty of the design is limited as it relies largely on well-known principles. However, the analysis yielding the logarithmic regret bound in this setting is important.

While the literature review is extensive, there are several other works that are worth including, such as:

A Lyapunov-Based Methodology for Constrained Optimization with Bandit Feedback
Semih Cayci, Yilin Zheng and Atilla Eryilmaz
Thirty-Sixth AAAI Conference on Artificial Intelligence, AAAI 2022

Budget-Constrained Bandits over General Cost and Reward Distributions
Semih Cayci, Atilla Eryilmaz, R. Srikant
International Conference on Artificial Intelligence and Statistics (AISTATS), PMLR 108, 2020.

The first of these works in particular also uses the lyapunov drift minimization techniques for constrained MABs, which is utilized in the PLLP algorithm design.

---

> ### Author Response · Authors · 2022-08-02
> **To Reviewer SVeo**
>
> Thanks for the positive comments and useful suggestions.
>
>
> ### Mentioned literature.
> The current version only includes the most related work in the introduction due to the page limit. The additional references suggested by the reviewer are indeed highly relevant and we appreciate them. We would be happy to include
> these related work in the full version of the paper.
>
> ### Detailed expansion of our examples.
> * Consider the crowdsourcing platform in which there are multiple heterogeneous workers to be hired for each arriving task. Each task needs a certain number of workers to complete, and each worker will be paid a fixed (possibly heterogeneous) fee for each contributed task. When the task is completed, it generates a random reward (or operation cost) to the platform, distribution of which may depend on the selected worker (i.e., the quality of worker differs).
> In practice, the participant ratio for various workers should also be guaranteed for fairness considerations. Therefore, the workers can be viewed as "arms", and the objective of the platform is to maximize the obtained reward subject to the fairness and price budget constraints.
>
> * Consider the scheduling problem of information gathering in Internet-of-Things (IoT) systems (e.g, aerial surveillance network, automated industrial plant), where a number of sensors continue to collect information (e.g., tire pressure, quantity of fuel) and transmit the data to the monitor.  Due to the limited (wireless) communication bandwidth, the monitor can only schedule a subset of sensors for transmission at each time. In practice, the monitor is usually required to satisfy some Quality of Service (QoS) constraints (e.g., throughput or mean-delay) on each source to ensure timely and efficient sampling and decision making.
> For example, for an autonomous driving car, data from obstacle/collision detection should be sent to the processor (monitor) instantly, while the sensor that measures the fuel availability can be updated in a lower frequency.
> To capture this attribute, the monitor could associate a minimum timely-throughput requirement with each sensor. Beyond the QoS constraints, each sensor also has a budget constraint of the power consumption due to limited battery capacity, as each transmission would consume a deterministic sensor-dependent power. The objective of the monitor is to accumulate the obtained information value subject to the QoS and (power) budget constraints for each sensor.

---

> > ### Comment · Reviewer_SVeo · 2022-08-08
> > **Most relevant work**
> >
> > Thank you for expanding the examples.
> >
> > Regarding the most relevant work, I believe both [Cayci et al. 2021] and
> >
> > Group-Fair Online Allocation in Continuous Time
> > Semih Cayci, Swati Gupta, Atilla Eryilmaz
> > Advances in Neural Information Processing Systems 33, NeurIPS 2020
> >
> > are quite relevant to this work for using similar ideas of implementing virtual-queues for managing constraints and analyzing their regret and violation performances. I am not sure what the authors mean by "full" version, but these deserve to be cited in this particular article, in my opinion.

---

> > > ### Author Response · Authors · 2022-08-09
> > > **To Reviewer SVeo**
> > >
> > > Thanks for your suggestions ! We agree that [Cayci et al. 2021] and [Cayci et al. 2020] are quite relevant to our work, and we just added them in the revision (see Appendix A).

---

### Official Review · Reviewer_rd83 · 2022-07-11

**Rating:** 6
**Confidence:** 4
**Soundness:** 3 good
**Presentation:** 3 good
**Contribution:** 3 good

**Summary:**

The authors study the Combinatorial bandits problem with linear constraints. This problem captures multiple existing problems including fairness constraints, and Knapsack constraints. The authors show that using UCB estimates for the mean reward of each arm in a constrained LP, and playing the arms according to the optimal solution of that LP ensures in expectation 0 constraint violation and logarithmic regret. As a special case, fairness constraints incurs constant regret when each arm is forced to choose a constant fraction of time. They augment this result with a Lagrangian based computationally efficient algorithm with $O(\sqrt{T})$ problem independent regret, and 0 constraint violation with high probability.

**Questions:**

Please compare the current work with the prior works mentioned in the weakness section.

**Limitations:**

Any negative societal impact for this work is hard to foresee.

**Strengths And Weaknesses:**

Strengths
- The proposed UCB-LP algorithm solves multiple well studied problems, such as semi-bandits with knapscak, and bandits with fairness constraints.
- The constant regret in fairness constraint is an interesting result (though not very surprising).
- The Lagrangian based optimization process (UCB-PLLP) is computationally efficient, and shows optimal minimax regret with no constraint violation with high probability.

Weakness
The paper is not well placed in the literature, in my opinion.
- The paper "The Combinatorial multi-armed bandit: General framework and applications." by Wei Chen et al.  already addresses the main challenge discussed in the paper, i.e. tackling exponential extreme points in the LP.  Similar ideas are used in a slightly different context in "Contextual Blocking Bandits" by Basu et al.
- The virtual queue based approach used in the UCB-PLLP is used in a series of work by H Yu and MJ Neely. E.g. "A Low Complexity Algorithm with $O(\sqrt{T})$ Regret and $O(1)$ Constraint Violations for Online Convex Optimization with Long Term Constraints" H Yu and MJ Neely.

Post Rebuttal: Upon discussing with the authors, I came to the conclusion that the current work indeed improves upon the technique used in  [Basu et al. 2021] for regret bounds in constrained LP with blocking, which was based on  [Chen et al., 2013] and [Wang and Chen, 2017]. The novelty in the virtual queue based technique was also made clear from discussions. The authors are requested to include these works and brief summary of discussions in the paper.

---

> ### Author Response · Authors · 2022-08-02
> **To Reviewer rd83**
>
> ### The challenge brought by exponentially many extreme points is already tackled by [Chen et al., 2013].
> We kindly point out that this is a total misunderstanding. Note that the notion of extreme points does not exist in [Chen et al., 2013] at all. Since [Chen et al., 2013] studies the unconstrained CMAB problem, in contrast to looking for an optimal sampling distribution over super-arms (i.e., optimal extreme point) in our work, their algorithm only needs to identify the best super-arm using UCB estimates of base arms and thus does not require solving an LP at every round. Therefore there is no LP, hence extreme points, in [Chen et al., 2013]. Obviously, as there is no super-arm that is universally optimal across all rounds but only an optimal sampling distribution over super-arms in our constrained CMAB setting, their analysis does not directly apply to our case.
> Furthermore, we remark that even in their setting of unconstrained CMAB problem (a special case of our setting), the technique they developed for handling the exponential number of super-arms is coarse and would incur a regret bound of $O(m^2 N\log T)$, which has a worsening multiplicative factor than ours. Our work could be seen as the non-trivial extension of theirs which both generalizes the setting to constrained CMAB and improves the regret bound's multiplicative factor at the same time.
>
>
> ### Comparison with [Basu et al., 2021]:
> [Basu et al., 2021] focuses on the (contextual) blocking bandit problem wherein once an arm is pulled it cannot be played again for a fixed number of consecutive rounds, and they also choose arms based on the sampling probability yielded by an LP whose constraints limit the frequency at which each arm can be selected.
> Here we argue that they did not face the difficulty discussed in our paper and their proof technique cannot directly apply to our situation. Specifically, in their model, the total number of arm-context pairs is only $N\cdot K$,
> where $N$ is the number of arms and $K$ is the number of contexts. Thus, they could maintain a cumulative reward counter for each arm-context pair and directly allocate the per-round regret to each arm-context pair.
> However, this is not the case in our problem since there is an exponential number of super-arms in the decision set. We address this difficulty by allocating per-round regret to our defined "dominant" base arms, which only incurs a increase on the regret bound by a factor of $2$.
>
>
> ### Comparison with [Yu and Neely, 2020]:
> [Yu and Neely, 2020] also used a virtual queue technique to track the problem of constrained online learning. However, we remark that their virtual queue based algorithm and analysis are only valid in the online convex optimization (OCO) setting, in which the full loss (reward) function at each round would be revealed after the decision making (we can think linear bandits with full-information feedback as a special case of OCO). While in our model, we do not have such observation due to the semi-bandit feedback. In fact, the update rule of virtual queue in our algorithm and theirs are totally different. Our virtual queue update rule is ${Q}(t+1)=\max (0,{Q}(t)+{g}({a}(t))+{\epsilon}_t\cdot 1_N)$, while theirs is ${Q}(t+1)=\max(-{g}({a}(t)) , {Q}(t)+{g}({a}(t)))$.
> In particular,
> our virtual queue technique involves a pessimism mechanism that adds a time-varying tightness constant to virtual queue update so that the (long-term) constraints could be guaranteed stringently with high-probability, while their algorithm can only achieve an $O(1)$ constraint violations.
>
>
> **We greatly appreciate the reviewer's comments. However, since the only listed weakness is misplacement in literature, we wonder if "borderline reject" is justified. We sincerely hope the reviewer will reconsider the rating based on our response.**
>
> ### References:
>
> [Chen et al., 2013]: The Combinatorial multi-armed bandit: General framework and applications
>
> [Basu et al., 2021]: Contextual Blocking Bandits
>
> [Yu and Neely, 2020]: A Low Complexity Algorithm with $O(\sqrt{T})$ Regret and $O(1)$ Constraint Violations for Online Convex Optimization with Long Term Constraints

---

> > ### Comment · Reviewer_rd83 · 2022-08-03
> > **More Clarification Needed**
> >
> > ## Chen et al., 2013 and Basu et al. 2021
> > In Basu et al. 2021 the constrained LP that is solved in each round is given as
> >
> > $max_z \sum_{i\leq N,j\leq K} \mu_{i,j} z_{i,j}, \text{ s.t. } \sum_{i} z_{i,j} \leq \alpha_j \forall j, \sum_{j} z_{i,j} \leq \beta_i \forall i, z_{i,j} \in [0,1] \forall i,j$.
> >
> > Hence, $N$ in this paper is equivalent to $NK$ in Basu et al. 2021. In other words, an arm-context pair in Basu et al. 2021 is equivalent to an arm in this paper.  Therefore, I do not understand the following comment
> >
> > >  Specifically, in their model, the total number of arm-context pairs is only $N \cdot K$, where $N$  is the number of arms and $K$ is the number of contexts ... However, this is not the case in our problem since there is an exponential number of super-arms in the decision set.
> >
> > One of the key tools used in Basu et al. 2021 to derive a logarithmic regret is taken from [Chen et al., 2013]. (If blocking is absent, as is the case in this paper, the tool in [Chen et al., 2013] is sufficient for regret bound in Basu et al. 2021).
> >
> > The authors should therefore provide a better explanation of why techniques in [Chen et al., 2013] is not useful here.
> >
> > ## Virtual Queue Based Technique
> > Thanks for the clarification on novelty in the virtual queue update proposed here vs Yu and Neely, 2020. Can you please comment on the novelty in virtual queue based method compared to Cayci et al. 2020, and Cayci et al. 2022  (mentioned by Reviewer SVeo)? That will help position this paper better.

---

> > > ### Author Response · Authors · 2022-08-06
> > > **Response to Reviewer rd83**
> > >
> > > ### Comparison with [Chen et al., 2013] and [Basu et al., 2021].
> > > Sorry that our response was not clear enough, here we provide a more detailed comparison of our work with these works.
> > >
> > > [Basu et al., 2021] utilizes techniques in [Chen et al., 2013] and  [Wang and Chen, 2017], where [Chen et al., 2013] proposes the general CMAB framework and [Wang and Chen, 2017] improves upon its regret bounds. However,
> > > we note that these techniques are not tight enough, and even applying the improved techniques in [Wang and Chen, 2017] to our setting results in regret bounds looser than Theorem 1. More specifically, the distribution dependent regret bound given in [Wang and Chen, 2017] is (in their notations) $\Theta(mB^2K\ln T/\Delta_{\min})$, where $B$ is the bounded smoothness constant, $m$ is the number of arms, and $K$ is the maximum number of arms that an action could trigger. With our notation and setting, the number of arms is $N$, and $K$ also becomes $N$ since our randomized policy ${x}(t)\in \mathcal{D}$ (LP solution) could have $N$ positive coordinates. So their bound is $\Theta(N^2\ln T/\Delta_{\min})$ with our notations, which is looser than our regret bound of $\Theta(mN\ln T/\Delta_{\min})$ since $m$ can be much smaller than $N$.
> > >
> > > The techniques used in [Basu et al., 2021] and [Wang and Chen, 2017] is to maintain a set of counters $\begin{Bmatrix}N_{i,j} \end{Bmatrix} $ for every arm $i$ and
> > >  allocate the regret to arms according to $N_{i,j}$. For every action (or extreme point) $S$, the support of $S$ is defined as $\hat{S}=\begin{Bmatrix}i\in[N] | p_i^{ S}> 0\end{Bmatrix}$, i.e., the set of arms that $S$ might trigger with positive probability (In our model, $|\hat{S}|=N$, while
> > > $|\hat{S}|$ is at most the sum of the number of arms and the number of contexts in [Basu et al., 2021]).
> > > The reason why their technique produces a looser bound is that, they set the regret allocated to every arm $i\in\hat{S}$ when action $S$ is played in round $t$ as (see Lemma 5 and the definition of $\kappa$ in appendix of [Wang and Chen, 2017]):
> > >
> > > $$\kappa_{j_i, T}(M_i, N_{i,j_i,t-1})=\begin{cases}
> > > 4\cdot 2^{-j_i}B&\text{if }N_{i,j_i,t-1}=0\\\\
> > > 2B\sqrt{\frac{72\cdot2^{-j_i}\ln T}{N_{i,j_i,t-1}}}&\text{if }1\leq N_{i,j_i,t-1}\leq l_{j_i,T}(M_i)\\\\
> > > 0&\text{if }N_{i,j_i,t-1}\geq l_{j_i,T}(M_i)+1
> > > \end{cases},
> > > $$
> > >
> > > where $M_i=\Delta_{\min}^i$ is the gap, and
> > > $l_{j_i,T}(M_i)=\lfloor\frac{288\cdot2^{-j_i}B^2K^2\ln T}{M_i}\rfloor.$
> > > So no regret is allocated to arm $i$ after the counter value $N_{i,j_i}$ exceeds $l_{j_i,T}(M_i)=\Theta(2^{-j_i} B^2K^2\ln T/\Delta_{\min})$. However, we note that (compared to our setting) the multiplicative factor $K^2$ is loose here. This bound is loose because all arms $i\in \hat{S}$ in the support of action $S$ are treated in the equal manner. More specifically, in equation (11) of the proof of lemma 5 in [Wang and Chen, 2017], they allocated the regret as
> > > \begin{equation}
> > > \Delta_{S_t}\leq B\sum_{i\in\hat{S_t}} p_i^{S_t} (\hat{\mu_i}(t)-\mu_i)\leq \cdots \leq 2B \sum_{i\in\hat{S_t}}  p_i^{S_t}(\hat{\mu_i}(t)-\mu_i)-\frac{M_{S_t}}{2B|S_t|}
> > > \end{equation}
> > >
> > > One can see that, the same term $\frac{M_{S_t}}{2B|S_t|}$ is attributed to every arm $i\in\hat{S_t}$ in the support. However, the arms in the support of action should not be associated with the same term, since their probability of being triggered by the action $S_t$ is different. This coarseness makes the final regret bound to scale with the size of the support, which is $K$ in their setting and $N$ in our setting, transforming the regret bound from our result of $\Theta(mN\ln T/\Delta_{\min})$ to $\Theta(N^2\ln T/\Delta_{\min})$.
> > >
> > > In contrast, our analysis is more fine-grained since we do not maintain a counter for every arm $i$ and allocate the regret according to the counter value, but allocate the regret to arm $i$ according to the number of times it is pulled, i.e., $h_i(t)$, directly. In line 198, we only allocate the regret to arms in $V(t):=\{i:h_i(t)\leq\frac{32m^2\ln t}{\Delta_{\min}^2}\}$, i.e., the "dominant arms" whose pulls is no more than $O(m^2\ln t/\Delta_{\min}^2)$ times.
> > > Note that the multiplicative factor $m^2$ here is tighter than the aforementioned $K^2$ factor (which is $N^2$ in our notation) since we bound arms in the support of action with a more smooth manner. More specifically, each arm is bounded according to $x_i(t)$, its probability of being triggered, and one no longer needs to distinguish arms in the support and other arms.
> > >
> > > Last but not least, we remark that the techniques from these works cannot establish a constant regret bound for the fairness setting. Our techniques for proving our constant regret bounds based on a combination of LP-sensitivity, martingale analysis, and the concentration inequality under a stochastic process that evolves over time.
> > >
> > > ### Reference
> > > [Wang and Chen, 2017]: Tighter regret bounds for influence maximization and other combinatorial semi-bandits with probabilistically triggered arms

---

> > > > ### Comment · Reviewer_rd83 · 2022-08-07
> > > > **The Response is Satisfactory**
> > > >
> > > > Thank you for your insightful response to my query.
> > > >
> > > > The response illustrates clearly how the current work adds to the literature. Can you please add a brief summary of this discussion to the related work? Furthermore, I feel, having the above details in the supplementary section will be quite helpful in providing an in-depth understanding of where the improvement comes from.
> > > >
> > > > The response to reviewer SVeo on the Virtual queue based technique is also satisfactory.

---

> > > > > ### Author Response · Authors · 2022-08-07
> > > > > **Thanks!**
> > > > >
> > > > > We sincerely thank the reviewer for reconsidering the score and the helpful comments. We will definitely incorporate these discussions in the revision.

---

> > > ### Author Response · Authors · 2022-08-06
> > > **Response to Reviewer rd83**
> > >
> > > ### Comparison with [Cayci et al., 2020].
> > > We remark that [Cayci et al., 2020] does not involve the virtual queue technique to handle the budget constraint as noted by reviewer SVeo. In fact, their work focused on the restrictive setting wherein only one resource is consumed over time and only one arm can be pulled at each round.
> > > Thus, there exists an optimal arm (whose expected reward-to-cost ratio is the highest) over the entire learning process, and the high-level idea of their algorithm is to identify this optimal arm.
> > >
> > > ### Comparison with [Cayci et al., 2021].
> > > Although [Cayci et al., 2021] also used a virtual queue technique to handle the budget constraints,
> > > our virtual queue update rule differs from theirs in the pessimistic mechanism via adding a time-varying tightness constant. Beyond the update rule of virtual queue, we also employ some new techniques in
> > > our Lyapunov analysis. For example,
> > > we establish a bound on the exponential moment
> > > of the virtual queue length (Lemma 6), which is the central focus of our high-probability guarantee on zero constraint violations. We also establish an upper bound on the $\epsilon_t$-tight term (incurred by our pessimistic mechanism) via comparing the optimal solution to the original LP problem and that to its $\epsilon_t$-tightened version based on LP-sensitivity (lemmas 3 and 4). These analysis techniques are not present in [Cayci et al., 2021].
> > >
> > > ### References:
> > > [Cayci et al., 2020]: Budget-Constrained Bandits over General Cost and Reward Distributions
> > >
> > > [Cayci et al., 2021]: A Lyapunov-Based Methodology for Constrained Optimization with Bandit Feedback

---

> > > > ### Comment · Reviewer_SVeo · 2022-08-08
> > > > **Correction and a further remark**
> > > >
> > > > The authors are correct in noting that the cited work [Cayci et al. 2020] is not relevant to this work. The correct citation should have been:
> > > >
> > > > Group-Fair Online Allocation in Continuous Time
> > > > Semih Cayci, Swati Gupta, Atilla Eryilmaz
> > > > Advances in Neural Information Processing Systems 33, NeurIPS 2020
> > > >
> > > > Both this work and the work [Cayci et al. 2021] sue virtual queues that are relevant prior work to this work. While there are differences and further improvements provided in this work (which is the reason behind my acceptance recommendation), it would be unfair not to include relevant work that use similar approaches.

---

### Official Review · Reviewer_7wPF · 2022-07-23

**Rating:** 6
**Confidence:** 2
**Soundness:** 2 fair
**Presentation:** 3 good
**Contribution:** 3 good

**Summary:**

This paper proposes the general bandit problem that subsumes Bandits with Knapsacks and Bandits with fairness constraints. UCB-LP, inspired by  SemiBwK algorithm [30], is proposed and its regret bound archives $O(mN \log T/\Delta_{\min})$. They also study the special class for fairness constraints and show that UCB-LP guarantees a constant regret. Finally, UCB-PLLP algorithm is proposed as a computational efficient algorithm and its regret bound archives $\tilde{O}(m \sqrt{T})$.


**Questions:**

Any new difficulties that arise from changing from a single-armed setting to a combinatorial (size constraint) setting?

Are there any lower bounds for fairness setting with respect to the $r_{\min}$?

===Post Rebuttal===

Thank you for the author feedback. I have read other reviews and discussion.

Especially, comparison with [Cayci et al., 2021]  and discussion on how this paper uses virtual queue techniques is clearer.
I hope the authors will incorporate them and position this paper in the literature.

So I will keep my score.

**Limitations:**

The paper is missing the section for conclusion and does not discuss any limitation or future direction. More discussion is appreciated.


**Strengths And Weaknesses:**

Strength:
1. Though UCB-LP is designed based on SemiBwK algorithm [30], generalization to linear constraints and regret analysis is done in a novel way. Naive analysis will suffer from the bound with size of extreme points of a feasible region of LP, i.e.,  $|\mathcal{B}|$ which is exponentially large. \


2. UCB-LP does not need prior knowledge while existing work [12, 29, 31] requires some knowledge of the problem instance, and UCB-LP reduces to the standard UCB-1 algorithm for the special case.



Weakness:

The whole procedure (pseudocode) for UCB-LP is missing in the main paper and supplemental material.

---

> ### Author Response · Authors · 2022-08-02
> **To Reviewer 7wPF**
>
>
>
> We thank the reviewer for the positive comment and respond to all stated concerns below.
>
> ### Pseudo-code of UCB-LP:
> We thank the reviewer for pointing this out.  We give a detailed description of UCB-LP in Section 2.2, thus, we omitted the pseudo-code of UCB-LP in the submitted draft due to space limit. We will add the pseudo-code to the paper in our final version.
>
> ### Technical difficulties when extending from single-armed setting to the combinatorial setting:
> **The main challenge is twofold.**
> * Note that in the single-armed setting, the total number of available actions is only $N$. Thus, we could maintain a cumulative reward counter for each base arm and directly allocate the per-round regret to the base arms.
> However, this is not the case for the combinatorial-setting since the number of actions (i.e, super-arm) that can be selected is exponentially large. Therefore, structural properties of the action set $\mathcal{A}$ must be utilized for obtaining a polynomial regret bound.
> We address this difficulty by the "regret allocation" technique (in line 188-196) that (approximately) allocates per-round regret to our defined "dominant" base arms, which only increases the regret bound by a constant factor of $2$.
>
>
> * Another challenge is how to obtain a feasible action $a(t)$ (i.e., $\sum_n a_n(t)\le m$) efficiently that maintains the marginal distribution of ${x}(t)$ (i.e., $\mathbb{P}[a_n(t)]=x_n(t)$) for $m>1$. Note that in the single-arm setting ($m=1$), this is trivial since ${x}(t)$ is directly the sampling probability vector over base arms, and the agent can easily choose ${a}(t)$ according to ${x}(t)$.
> To address this algorithmic challenge, we choose ${a}(t)$ according to a sample rule constructed from the convex decomposition of ${x}(t)$, and we design a computationally-efficient convex decomposition scheme for ${x}(t)$ stated in Appendix A.1.
>
> ### Lower bounds for fairness setting with respect to $r_{\min}$:
> This is an interesting problem and we thank the reviewer for proposing it. We provided an argument for this in line 279-283 in an asymptotic way by showing that the regret is no longer constant in $T$ when $r_{\min}$ tends to zero. However, we do not know an exact lower bound for this problem (the exploration of exact lower bound requires the technique from information theory) and we take this problem as an interesting topic for future study.
>
> ### More discussion about conclusion, limitation and future direction is appreciated:
> As stated above, we think pursuing a lower bound for the fairness setting is an interesting future direction. We thank the reviewer again for proposing such a question and will add these discussions to the paper.

---

> ### Author Response · Authors · 2022-08-10
> **To Reviewer 7wPF**
>
> Although hope we have addressed all of your main concerns, any further comments or questions are welcomed! Thank you for your time!

---

### Author Response · Authors · 2022-08-09
**Follow-up**

Dear Reviewers,

We hope that our answers comforted your opinion on the paper. Even if the discussion period ends soon, we would be glad to answer any additional question.

Best, the authors.

---

### Meta-Review · Area_Chair_kpJV · 2022-08-21

**Recommendation:** Accept
**Confidence:** Certain

**Metareview:**

We thank the authors for their submission.

This work presents a novel framework for bandits with long-term constraints, generalizing both bandits with knapsacks and bandits with fairness constraints. The authors present efficient no-regret algorithms as well as remove some problem-specific assumptions made by prior work.

**Award:**

No

---

### Decision · Program_Chairs · 2022-09-14

Accept